# A Game-Theoretic Framework for Measuring and Explaining Metric Compatibility in Fair Machine Learning

**Lingfeng Zhang** [1]   **Jingran Yang** [1]   **Zhaohui Wang** [1]   **Min Zhang** [1]   **Qing Zhang** [1]

## Abstract

Machine learning fairness research documents trade-offs but lacks quantitative frameworks to measure intrinsic metric compatibility without requiring causal graphs. We introduce a game-theoretic framework that decomposes metrics into interaction vectors, enabling compatibility measurement between metrics via cosine similarity and mechanistic attribution to attribute coalitions. Through analysis of 6 datasets, 7 models, and 6 debiasing methods, we reveal that fairness and utility are often structurally orthogonal (median compatibility $\approx 0$) rather than diametrically opposed, with conflicts driven by sparse, low-order interactions. We further show that debiasing improves fairness by compressing the compatibility space—reducing compatibility of both synergistic and conflicting relationships—rather than eliminating conflicts, providing a mechanistic basis for understanding metric alignment.

## 1. Introduction

Machine Learning (ML) systems increasingly mediate high-stakes decisions in criminal justice, employment, and lending (ProPublica, 2016; Chan & Wang, 2018; Liu et al., 2018), raising critical concerns about algorithmic fairness. A foundational challenge is the tension between fairness and utility: improving one objective often degrades another (Dwork et al., 2012). While impossibility theorems establish qualitative incompatibilities (Chouldechova, 2017; Kleinberg et al., 2017; Pleiss et al., 2017) and empirical studies trace Pareto frontiers (Agarwal et al., 2018; Taufiq et al., 2024), existing work often treats trade-offs as binary phenomena without quantifying their *degree* or explaining their *mechanisms*.

---

[1]Software Engineering Institute, East China Normal University, Shanghai, China. Correspondence to: Min Zhang <mzhang@sei.ecnu.edu.cn>.

*Proceedings of the $43^{rd}$ International Conference on Machine Learning*, Seoul, South Korea. PMLR 306, 2026. Copyright 2026 by the author(s).

Current approaches face three key limitations. First, impossibility results certify incompatibility at the boundary conditions of feasibility, but do not characterize the *geometry of the optimization landscape* before those boundaries are reached—specifically, the degree of conflict in continuous optimization (Chouldechova, 2017). Second, extensive benchmarks quantify accuracy loss (Hort et al., 2021; Chen et al., 2023b; Taufiq et al., 2024) but treat the underlying drivers as a black box, lacking mechanistic attribution. Third, causal frameworks (Kusner et al., 2017; Plecko & Bareinboim, 2025) attribute trade-offs to data bias but rely on strong assumptions (e.g., causal graphs), limiting applicability. Recent work shows fairness constraints can improve accuracy under certain bias models (Wick et al., 2019; Sharma & Deshpande, 2024), but conditions are restrictive. These gaps leave practitioners without principled guidance on *when* and *why* trade-offs emerge.

We address these limitations by introducing a game-theoretic framework that quantifies and explains metric compatibility. While prior work (Zhang et al., 2025) applies Harsanyi interactions to decompose group fairness metrics for bias mitigation, we extend this to a *unified analysis of relationships* between multiple fairness and utility metrics. Specifically, we decompose each metric into a vector of attribute-interaction contributions via Harsanyi dividends (Harsanyi, 1982), and define compatibility as the alignment between these vectors. We define *compatibility* as the cosine similarity between these vectors, enabling mechanistic attribution to specific attribute coalitions without assuming causal graphs.

Our empirical study spans 6 datasets, 7 models, 6 debiasing methods, and 9 metrics covering group fairness (Calders et al., 2009; Hardt et al., 2016), individual fairness (Dwork et al., 2012; Galhotra et al., 2017), and utility. We make four key contributions: **(1)** Fairness and utility metrics predominantly exhibit structural orthogonality (median $\mathcal{C} \approx 0$) rather than diametric opposition, helping explain why modest empirical trade-offs can occur in some settings. **(2)** Sparse, low-order coalitions involving sensitive attributes drive compatibility, with the empty coalition capturing dataset imbalance—formalizing prior observations (Kamiran & Calders, 2012) without causal graphs. **(3)** Debi-

*Table 1.* Machine learning fairness metrics. In the formulations, $A$ represents the set of protected attribute(s), $S$ is the set of value combinations of the attribute(s) in $A$, $Y$ denotes the ground truth label, $\hat{Y}$ denotes the predicted label, $x'$ represents a similar sample to $x$, $ns$ denotes an assignment of non-sensitive attributes, $s$ denotes an assignment of sensitive attributes, 1 is the favorable class, 0 is the unfavorable class, and $d(\cdot)$ is a similarity function.

| METRIC | DEFINITION |
|---|---|
| SPD | $\max_{s \in S} P[\hat{Y} = 1 \mid A = s] - \min_{s \in S} P[\hat{Y} = 1 \mid A = s]$ |
| EOD | $\max_{s \in S} P[\hat{Y} = 1 \mid A = s, Y = 1] - \min_{s \in S} P[\hat{Y} = 1 \mid A = s, Y = 1]$ |
| PED | $\max_{s \in S} P[\hat{Y} = 1 \mid A = s, Y = 0] - \min_{s \in S} P[\hat{Y} = 1 \mid A = s, Y = 0]$ |
| AOD | $\frac{1}{2}\left[ \begin{array}{l} \max_{s \in S}(P[\hat{Y} = 1 \mid A = s, Y = 0] + P[\hat{Y} = 1 \mid A = s, Y = 1]) \\ -\min_{s \in S}(P[\hat{Y} = 1 \mid A = s, Y = 0] + P[\hat{Y} = 1 \mid A = s, Y = 1]) \end{array} \right]$ |
| GIFVR | $P_x[\exists x' : d(x, x') = 1 \wedge \hat{Y}(x) \neq \hat{Y}(x')]$ |
| CFVR | $P_{x=(ns,s)}[\exists x' = (ns', s') : ns = ns' \wedge s \neq s' \wedge \hat{Y}(x) \neq \hat{Y}(x')]$ |

*Table 2.* Machine learning performance metrics. $TP, TN, FP,$ and $FN$ denote the number of True Positive, True Negative, False Positive, and False Negative samples, respectively.

| METRIC | DEFINITION |
|---|---|
| ACCURACY | $(TP + TN)/(TP + FP + TN + FN)$ |
| PRECISION | $TP/(TP + FP)$ |
| RECALL | $TP/(TP + FN)$ |
| FPR | $FP/(FP + TN)$ |
| F1-SCORE | $(2 \times \text{PRECISION} \times \text{RECALL})/(\text{PRECISION} + \text{RECALL})$ |
| MCC | $\frac{TP \times TN - FP \times FN}{\sqrt{(TP+FP)(TP+FN)(TN+FP)(TN+FN)}}$ |
| ROC-AUC | $\int_0^1 \text{RECALL}\, d(\text{FPR})$ |

asing compresses compatibility space (reducing both synergies and conflicts), with aggressive methods (Fair-SMOTE: 44.2% reduction) sacrificing more alignment than gentle methods (MAAT: 11.87%). **(4)** Compatibility is highly context-specific (predictive models fail with negative $R^2$), necessitating empirical evaluation per application (Sec. 4.3, Appendix B.3, B.6). Unlike prior work proving impossibility (Chouldechova, 2017) or mapping Pareto curves (Taufiq et al., 2024), we explain *why* metrics conflict through interpretable interactions.

## 2. Background

### 2.1. Metrics

We present commonly used fairness metrics in Table 1, where group fairness metrics focus on the statistical parity of different demographic groups, including Statistical Parity Difference (SPD) (Calders et al., 2009; Chen et al., 2024), Equal Opportunity Difference (EOD) (Hardt et al., 2016; Chen et al., 2024), Predictive Equality Difference

(PED) (Hardt et al., 2016), and Average Odds Difference (AOD) (Hardt et al., 2016; Chen et al., 2024). Individual fairness metrics require that similar individuals be treated similarly, including Global Individual Fairness Violation Ratio (GIFVR) (Dwork et al., 2012; Khedr & Shoukry, 2023) and Causal Fairness Violation Ratio (CFVR) (Galhotra et al., 2017). Such definitions are generalizable to single or multiple sensitive attributes. We present commonly used machine learning utility metrics in Table 2, including accuracy, precision, recall, False Positive Rate (FPR), F1-score, Matthews Correlation Coefficient (MCC), and ROC-AUC.

### 2.2. Fairness-Utility Trade-offs

The tension between algorithmic fairness and utility is a foundational issue in ML. Research on this topic has largely progressed along three fronts, each with distinct limitations that motivate our work.

**Impossibility theorems** prove that satisfying calibration alongside equalized error rates is impossible except in trivial cases (Chouldechova, 2017; Kleinberg et al., 2017; Pleiss et al., 2017), but do not quantify the cost in continuous optimization. **Pareto frontiers** (Kamishima et al., 2011; Hardt et al., 2016; Agarwal et al., 2018; Tang et al., 2023; Fallah et al., 2025) characterize trade-offs empirically, confirming data-dependence (Hort et al., 2021; Chen et al., 2023b; Taufiq et al., 2024), yet treat underlying drivers as a black box. **Causal frameworks** (Kusner et al., 2017; Nabi & Shpitser, 2018; Ji et al., 2023; Plecko & Bareinboim, 2025; Wick et al., 2019; Sharma & Deshpande, 2024; Leininger et al., 2025) attribute trade-offs to data bias but require strong assumptions (causal graphs, unbiased labels), limiting applicability.

### 2.3. Game-Theoretic Interactions in Explainable Artificial Intelligence

The Shapley value framework (Lundberg & Lee, 2017), refined for feature dependence (Aas et al., 2021), has evolved from simple attribution to capturing complex multivariate interactions (Zhang et al., 2021). Recent theoretical breakthroughs demonstrate that DNN inference can be faithfully disentangled into sparse, symbolic interaction primitives (Ren et al., 2023a; 2024), which serve as trustworthy knowledge representations (Li & Zhang, 2023). In the fairness domain, HIFI (Zhang et al., 2025) pioneered the use of Harsanyi interactions to decompose group fairness metrics, revealing how sensitive information is implicitly encoded. Building on HIFI's decomposition mechanism, this work expands the scope to a broader set of utility and fairness objectives, and further captures intrinsic metric relationships.

**Relationship to HIFI.** Our work extends HIFI (Zhang et al., 2025) in three dimensions: **(1) Scope:** from single group fairness metrics to diverse group, individual, and utility met-

rics; **(2) Inter-Metric Analysis:** introducing metric compatibility to analyze pairwise relationships, revealing synergies and trade-offs; **(3) Optimization Dynamics:** connecting compatibility to debiasing dynamics (Sec. 4.3) and Pareto frontiers (Sec. 4.4). While HIFI asks "*why is this model biased?*", we ask "*why do metrics conflict, and how does debiasing reshape relationships?*"

### 2.4. Problem Formulation and Positioning

**Problem Formulation.** We focus on binary classification where a model $v(\cdot)$ predicts $P[\hat{Y} = 1|x]$ given an input $x$ comprising non-sensitive attributes and sensitive attributes (not necessarily binary). We analyze a set of metrics $\mathcal{M}$ covering group fairness, individual fairness, and utility. Unlike prior work that measures trade-offs via multiple post-optimization outcomes (e.g., Pareto frontiers), our goal is to quantify the *intrinsic* compatibility between any pair of metrics $(m_1, m_2)$—defined as the structural alignment of their requirements on $v$—and attribute this alignment to specific attribute interactions without relying on causal graphs. We emphasize that compatibility does not replace Pareto analysis; rather, it provides a complementary, structural view that can help interpret and anticipate the trade-offs observed on the Pareto frontier (as demonstrated in Sec. 4.4).

**Our Positioning.** We propose a game-theoretic framework that leverages Harsanyi interactions to decompose each metric $m \in \mathcal{M}$ into a representation vector of attribute-interaction contributions, and define compatibility as the cosine similarity between these vectors. This approach bridges the gap between descriptive benchmarks and causal mechanisms (Sec. 2.2):

**Mechanistic yet Oracle-Free:** We explain inter-metric compatibility drivers without assuming causal graphs or unbiased labels. **Quantitative and Interpretable:** We characterize relationships with bounded values in $[-1, 1]$ and decompose compatibility into coalition-level contributions. **Unified Multi-Metric Analysis:** Unlike HIFI (Zhang et al., 2025), we extend interaction-based decomposition to a diverse set of metrics, analyze pairwise relationships, and connect them to debiasing dynamics and Pareto frontiers.

## 3. Methodology

In this section, we present our framework for quantifying and explaining the intrinsic compatibility between metrics. Since standard metrics are often discrete, we first formulate continuous approximations to facilitate fine-grained analysis (Sec. 3.1). Leveraging these proxies, we apply game-theoretic decomposition to resolve each metric into a vector of attribute-level interactions (Sec. 3.2). Based on this decomposition, we formally define compatibility as the alignment between interaction vectors (Sec. 3.3).

*Table 3.* Continuous approximations of nine metrics based on predicted probabilities $v(x) = P[\hat{Y} = 1|x]$. Here, $P_p$ and $P_{up}$ denote distributions of privileged and unprivileged groups; $P_f$ and $P_{uf}$ denote distributions of samples with favorable ($Y = 1$) and unfavorable ($Y = 0$) ground truth labels. Combinations like $P_{p\&f}$ represent the intersection of these groups. $\alpha$ is the base rate $P[Y = 1]$.

| METRIC | DEFINITION |
|---|---|
| $\widetilde{\text{SPD}}$ | $\mathbb{E}_{x \sim P_p} v(x) - \mathbb{E}_{x \sim P_{up}} v(x)$ |
| $\widetilde{\text{EOD}}$ | $\mathbb{E}_{x \sim P_{p\&f}} v(x) - \mathbb{E}_{x \sim P_{up\&f}} v(x)$ |
| $\widetilde{\text{PED}}$ | $\mathbb{E}_{x \sim P_{p\&uf}} v(x) - \mathbb{E}_{x \sim P_{up\&uf}} v(x)$ |
| $\widetilde{\text{AOD}}$ | $\frac{1}{2} \left[ \begin{array}{c} (\mathbb{E}_{x \sim P_{p\&f}} v(x) + \mathbb{E}_{x \sim P_{p\&uf}} v(x)) \\ - (\mathbb{E}_{x \sim P_{up\&f}} v(x) + \mathbb{E}_{x \sim P_{up\&uf}} v(x)) \end{array} \right]$ |
| $\widetilde{\text{GIFVR}}$ | $\mathbb{E}_x \left[ \max_{x':d(x,x')=1} v(x') - \min_{x':d(x,x')=1} v(x') \right]$ |
| $\widetilde{\text{CFVR}}$ | $\mathbb{E}_{ns} \max_{s,s' \in S} [v(ns, s) - v(ns, s')]$ |
| $\widetilde{\text{ACCURACY}}$ | $\alpha \cdot \mathbb{E}_{x \sim P_f} v(x) + (1 - \alpha) \cdot \mathbb{E}_{x \sim P_{uf}} [1 - v(x)]$ |
| $\widetilde{\text{RECALL}}$ | $\mathbb{E}_{x \sim P_f} v(x)$ |
| $\widetilde{\text{FPR}}$ | $\mathbb{E}_{x \sim P_{uf}} v(x)$ |

### 3.1. Continuous Metric Approximation

Standard fairness and utility metrics (Tables 1-2) are defined on discrete predictions $\hat{Y} \in \{0, 1\}$, which are unsuitable for fine-grained interaction analysis. While our framework cannot decompose the *original discrete metrics* directly, we note that model training optimizes continuous loss functions (e.g., cross-entropy) rather than discrete metrics. Therefore, analyzing continuous proxies based on predicted probabilities $v(x) = P[\hat{Y} = 1|x]$ reflects the *optimization landscape* that governs model behavior and captures implicit biases overlooked by standard metrics (Wang et al., 2025).

Table 3 presents these continuous approximations. We specifically focus on metrics expressible as linear expectations of sample predictions, which enable additive decomposition via Harsanyi interactions. Complex metrics like precision, F1-score, MCC, and ROC-AUC are excluded as they involve non-linear operations (e.g., ratios of expectations) that hinder such decomposition. For included metrics, we define distributions by group membership and ground truth labels. For instance, $\widetilde{\text{SPD}}$ measures the difference in expected prediction confidence between privileged and unprivileged groups.

To validate the reliability of these proxies, we analyzed their correlation with standard metrics across 36 experimental configurations (6 datasets × 6 methods, aggregated over 7 models × 10 seeds). Crucially, to emulate the computational constraints of the subsequent interaction analysis, we evaluated our proxies using a random subsample of 500 test

*Table 4.* Correlation coefficients between original discrete metrics (computed on full test sets) and continuous proxies (evaluated on 500 random samples to emulate the computational constraints of interaction analysis). All correlations are significant at $p < 0.05$. Strong to excellent alignment validates the fidelity of our proxies despite sampling variance.

| CORRELATION | SPD | EOD | PED | AOD | GIFVR | CFVR | ACCURACY | RECALL | FPR |
|---|---|---|---|---|---|---|---|---|---|
| KENDALL $\tau$ | 0.6952* | 0.5873* | 0.7556* | 0.6095* | 0.8413* | 0.9187* | 0.8562* | 0.4952* | 0.8921* |
| SPEARMAN $\rho$ | 0.8505* | 0.7429* | 0.8994* | 0.7645* | 0.9606* | 0.9806* | 0.9631* | 0.6700* | 0.9786* |

instances (or the full set if smaller), while actual metrics were calculated on the full test set. As shown in Table 4, our proxies exhibit strong to excellent alignment with actual metrics despite this sampling variance. Utility metrics like FPR ($\rho = 0.979$) and accuracy ($\rho = 0.963$), as well as individual fairness metrics like CFVR ($\rho = 0.981$), show near-perfect Spearman correlations. The results confirm that our continuous proxies are robust to sampling noise and faithfully represent the original metrics, providing a reliable foundation for efficient game-theoretic analysis.

For non-linear metrics such as precision, F1-score, MCC, and ROC-AUC, our current framework should be used as a mechanistic proxy diagnosis rather than a direct decomposition of the final industrial objective. In practice, one can first apply compatibility analysis to decomposable linear components or proxy metrics to identify conflict-driving coalitions, and then validate the selected model using the target non-linear metrics on a held-out set.

### 3.2. Game-Theoretic Decomposition

**Harsanyi Interaction.** To dissect the driving forces behind each metric, we leverage the Harsanyi interaction (Harsanyi, 1982), a game-theoretic concept that uniquely decomposes the model output $v(x)$ into contributions from different attribute coalitions. Let $N$ be the set of all input attributes. For any coalition $C \subseteq N$, the Harsanyi interaction $I_C(x)$ quantifies the marginal contribution of the specific pattern formed by attributes in $C$, modeled as an AND relationship (Ren et al., 2023a). Formally:

$$I_C(x) = \sum_{C' \subseteq C} (-1)^{|C|-|C'|} \cdot v(x_{C'}), \quad (1)$$

where $x_{C'}$ is the input $x$ with attributes in $N \setminus C'$ masked to their baseline values (Ren et al., 2023b). A crucial property is *faithfulness*: the sum of interactions perfectly recovers the model output, i.e., $v(x) = \sum_{C \subseteq N} I_C(x)$. This additivity allows us to attribute the model's behavior directly to specific attribute combinations.

**Metric Decomposition.** Since our continuous proxies (Table 3) are linear expectations of $v(x)$, we can substitute $v(x)$ with its interaction sum to decompose each metric. Taking

$\widetilde{\text{SPD}}$ as an example:

$$\begin{aligned} \widetilde{\text{SPD}} &\triangleq \mathbb{E}_{x \sim P_p} v(x) - \mathbb{E}_{x \sim P_{up}} v(x) \\ &= \sum_{C \subseteq N} \left[ \mathbb{E}_{x \sim P_p} I_C(x) - \mathbb{E}_{x \sim P_{up}} I_C(x) \right]. \end{aligned} \quad (2)$$

Based on this linearity, we formally define the representation used for subsequent compatibility analysis:

**Definition 3.1** (Metric Interaction Vector). For any metric $m$, its structural dependence on input attributes is characterized by a vector $\mathbf{I}_m \in \mathbb{R}^{2^{|N|}}$, where the component $\mathbf{I}_m(C)$ represents the contribution of coalition $C \subseteq N$ to the metric value, such that $m = \sum_{C \subseteq N} \mathbf{I}_m(C)$.

For $\widetilde{\text{SPD}}$, the component is $\mathbf{I}_{\text{SPD}}(C) = \mathbb{E}_{x \sim P_p} I_C(x) - \mathbb{E}_{x \sim P_{up}} I_C(x)$. A positive value indicates that coalition $C$ pushes the model to favor the privileged group. This strategy applies to all metrics (summarized in Table 5 and derived in Appendix A.2). For utility metrics, to ensure consistency with fairness metrics (lower values are preferred), we decompose their error forms: $\widetilde{\text{Accuracy}_\downarrow} = 1 - \widetilde{\text{Accuracy}}$ and $\widetilde{\text{Recall}_\downarrow} = 1 - \widetilde{\text{Recall}}$. In this way, the discrepancy in optimization directions need not be considered in the following computation of metric compatibility.

Notably, for Causal Fairness, our decomposition reveals a structural property aligning with the intuition of causal discrimination (Galhotra et al., 2017):

**Proposition 3.2** (Sparsity of Causal Fairness). *Under the assumption of global baseline values (e.g., zero or mean), the interaction component $\mathbf{I}_{CFVR}(C)$ is non-zero **only if** the coalition $C$ contains at least one sensitive attribute (i.e., $C \cap A \neq \emptyset$). Conversely, for any coalition consisting solely of non-sensitive attributes ($C \subseteq N \setminus A$), $\mathbf{I}_{CFVR}(C) = 0$.*

*Proof.* When $C$ contains only non-sensitive attributes, the sensitive attributes are always masked to the same baseline values in both $v(ns, s_p)$ and $v(ns, s_{up})$, making their Harsanyi interactions identical. See Appendix A.3 for the full derivation. □

This proposition shows that, under any globally fixed baseline, the CFVR decomposition isolates coalitions that include sensitive attributes. The result should not be interpreted as a causal claim about real-world interventions; it is

*Table 5.* The component definition $\mathbf{I}_m(C)$ for each metric vector. Note that for $\widetilde{\text{Acc}}$ and $\widetilde{\text{Recall}}$, we decompose their complements. $x^+ = argmax_{x':d(x,x')=1}v(x')$, $x^- = argmin_{x':d(x,x')=1}v(x')$, $s_p = argmax_{s \in S}v(ns, s)$, $s_{up} = argmin_{s \in S}v(ns, s)$, $\alpha = P(Y = 1)$, $\mathbf{I}_{\text{Accuracy}_\downarrow}(\emptyset)$ additionally includes the constant $\alpha$, and $\mathbf{I}_{\text{Recall}_\downarrow}(\emptyset)$ additionally includes the constant 1.

| METRIC VECTOR | COMPONENT $\mathbf{I}_m(C)$ |
|---|---|
| $\mathbf{I}_{\text{SPD}}$ | $\mathbb{E}_{x \sim P_p} I_C(x) - \mathbb{E}_{x \sim P_{up}} I_C(x)$ |
| $\mathbf{I}_{\text{EOD}}$ | $\mathbb{E}_{x \sim P_{p\&f}} I_C(x) - \mathbb{E}_{x \sim P_{up\&f}} I_C(x)$ |
| $\mathbf{I}_{\text{PED}}$ | $\mathbb{E}_{x \sim P_{p\&uf}} I_C(x) - \mathbb{E}_{x \sim P_{up\&uf}} I_C(x)$ |
| $\mathbf{I}_{\text{AOD}}$ | $\frac{1}{2}(\mathbf{I}_{\text{EOD}}(C) + \mathbf{I}_{\text{PED}}(C))$ |
| $\mathbf{I}_{\text{GIFVR}}$ | $\mathbb{E}_x[I_C(x^+) - I_C(x^-)]$ |
| $\mathbf{I}_{\text{CFVR}}$ | $\mathbb{E}_{ns}[I_C(ns, s_p) - I_C(ns, s_{up})]$ |
| $\mathbf{I}_{\text{ACCURACY}_\downarrow}$ | $(1 - \alpha) \cdot \mathbb{E}_{x \sim P_{uf}} I_C(x) - \alpha \cdot \mathbb{E}_{x \sim P_f} I_C(x)$ |
| $\mathbf{I}_{\text{RECALL}_\downarrow}$ | $-\mathbb{E}_{x \sim P_f} I_C(x)$ |
| $\mathbf{I}_{\text{FPR}}$ | $\mathbb{E}_{x \sim P_{uf}} I_C(x)$ |

a structural property of the masking-based decomposition. As discussed in Appendix B.8 and Appendix B.9, the numerical interaction values can vary with the masking strategy, although our main compatibility patterns are stable across several alternatives.

### 3.3. Compatibility Measurement

With the metric interaction vectors $\mathbf{I}_m$ derived in Definition 3.1, we can now quantify the intrinsic relationship between any pair of metrics. Since we have unified all metrics such that lower values are preferred (using error forms for utility), *compatibility* is defined as the structural alignment between their interaction vectors.

We employ Cosine Similarity as the measure of compatibility. This choice is motivated by the fact that we are interested in whether two metrics are driven by the same *underlying mechanisms* (i.e., the direction of the vector), rather than their absolute magnitudes.

**Why cosine similarity?** We use cosine similarity because a compatibility measure should satisfy three requirements in fairness-critical diagnosis. First, it should be *bounded and sign-interpretable*, so that positive, near-zero, and negative values correspond to synergy, orthogonality, and conflict. Second, it should be *scale-invariant*, since we aim to compare whether two metrics depend on the same coalition-level mechanisms rather than whether their raw magnitudes are similar. Third, it should be *decomposable over coalitions*, enabling attribution of the final compatibility score to individual feature coalitions as in Eq. 3. These requirements

motivate cosine similarity over unnormalized or rank-only alternatives.

**Definition 3.3** (Metric Compatibility). The compatibility between two metrics $m_1$ and $m_2$, denoted as $\mathcal{C}(m_1, m_2)$, is the cosine similarity between their corresponding interaction vectors:

$$\mathcal{C}(m_1, m_2) \triangleq \frac{\mathbf{I}_{m_1} \cdot \mathbf{I}_{m_2}}{|\mathbf{I}_{m_1}||\mathbf{I}_{m_2}|} = \sum_{C \subseteq N} \frac{\mathbf{I}_{m_1}(C)\mathbf{I}_{m_2}(C)}{|\mathbf{I}_{m_1}||\mathbf{I}_{m_2}|}. \quad (3)$$

**Geometric Interpretation.** The value $\mathcal{C}(m_1, m_2) \in [-1, 1]$ quantifies angular alignment: **Synergy** ($\mathcal{C} > 0$) implies metrics are driven by similar patterns in the same direction; **Trade-off** ($\mathcal{C} < 0$) indicates shared but opposing mechanisms; and **Orthogonality** ($\mathcal{C} \approx 0$) suggests metrics rely on disjoint attribute interactions, allowing quasi-independent optimization.

**Coalition-Level Attribution.** Beyond a single scalar score, our framework enables fine-grained diagnosis of *why* two metrics are compatible or conflicting. The term

$$\text{Contrib}(C; m_1, m_2) = \frac{\mathbf{I}_{m_1}(C)\mathbf{I}_{m_2}(C)}{|\mathbf{I}_{m_1}||\mathbf{I}_{m_2}|} \quad (4)$$

represents the contribution of coalition $C$ to the overall compatibility $\mathcal{C}(m_1, m_2) = \sum_C \text{Contrib}(C; m_1, m_2)$. For instance, if $\mathcal{C}(m_1, m_2) < 0$ (trade-off), we can identify coalitions with large negative products $\mathbf{I}_{m_1}(C)\mathbf{I}_{m_2}(C) \ll 0$ as the primary drivers of conflict. This attribution reveals whether conflicts stem from sensitive attributes ($C \cap A \neq \emptyset$), non-sensitive features, or dataset-level biases (captured by $C = \emptyset$), providing actionable insights for targeted interventions (e.g., preprocessing).

## 4. Empirical Results

We evaluate on six datasets (Becker & Kohavi, 1996; da Silva, 2019; ProPublica, 2016; Kahn, 1994; Moro et al., 2014; Jánosi et al., 1989), seven models (LR, SVM, DT, RF, MLP, XGBoost (Chen & Guestrin, 2016), TabNet (Arik & Pfister, 2021)), and six debiasing methods (Kamiran & Calders, 2012; Li et al., 2023; Dwork et al., 2012; Yang et al., 2024; Chakraborty et al., 2021; Chen et al., 2022; Zhang et al., 2025). All configurations repeated 10 times (2,700+ instances)[1]. We sample 500 test instances for efficiency. Details and more experiments are listed in Appendix B.

### 4.1. Compatibility Landscape

We evaluate the pairwise compatibility $\mathcal{C}(m_1, m_2)$ across 36 metric pairs using vanilla models, aggregating results from 420 experimental configurations. Figure 1 visualizes

---

[1]Code available at https://github.com/LingfengZhang98/Metric-Compatibility.

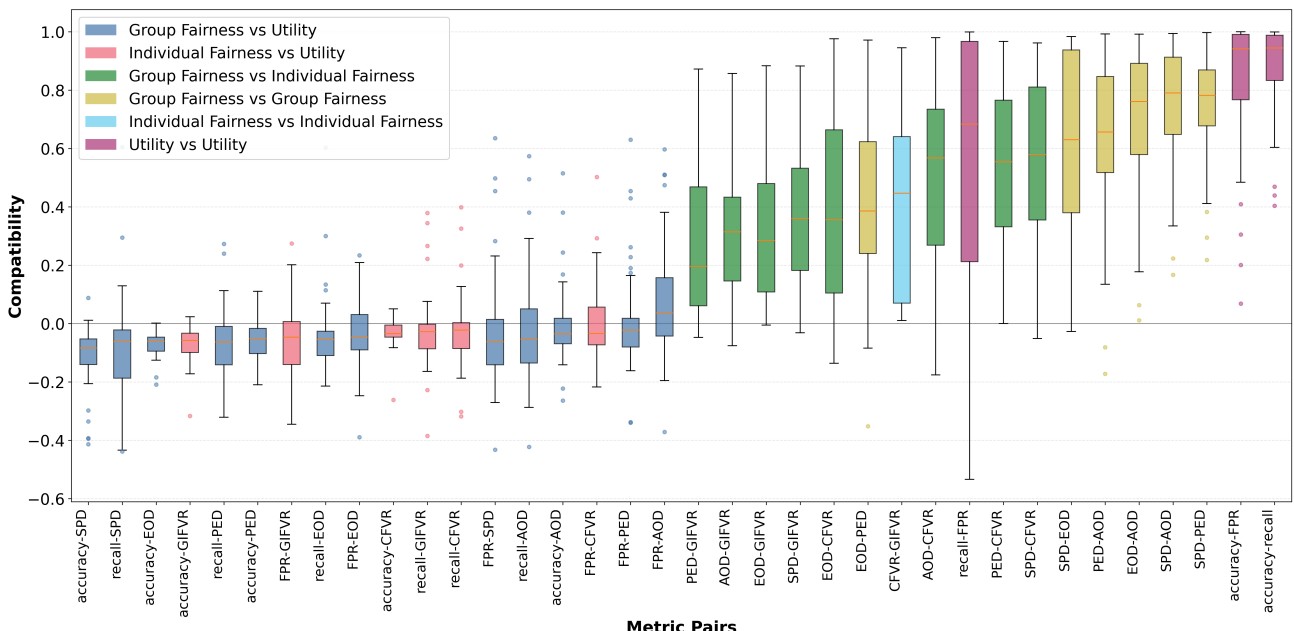

*Figure 1.* Global compatibility landscape across 36 metric pairs evaluated on vanilla models. Each boxplot aggregates 420 data points (6 datasets × 7 models × 10 seeds). Pairs are sorted by mean compatibility and colored by category. *Key observations:* (1) Utility-Utility pairs exhibit near-perfect alignment ($\mathcal{C} > 0.9$); (2) Group-Group and Individual-Individual fairness pairs show strong internal synergy; (3) Fairness-Utility pairs predominantly cluster near zero (orthogonality) rather than $-1$ (diametric opposition), with median $\mathcal{C} \in [-0.1, 0]$. This geometric structure helps explain why fairness-utility trade-offs are often milder than diametric opposition would suggest.

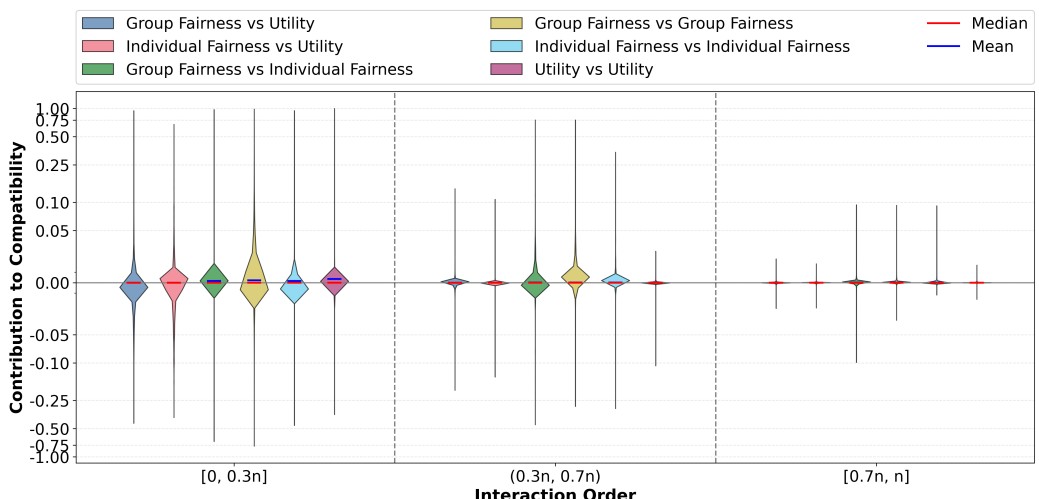

*Figure 2.* Coalition contributions to compatibility by interaction order. Violin plots show Contrib($C; m_1, m_2$) density for orders $|C|$ across categories ($n = |N|$ features). *Key observations:* (1) **Sparsity:** most coalitions $\approx 0$; (2) **Low-order dominance:** variance decreases with order—compatibility driven by simple patterns (e.g., single attributes/pairs).

this landscape, revealing a distinct hierarchical structure that challenges the presumed sharp opposition between fairness and utility:

**Hierarchical Compatibility Structure.** Figure 1 shows clear hierarchy: Utility-Utility pairs exhibit near-perfect alignment, followed by Group-Group and Individual-Individual synergy. Group-Individual pairs show positive alignment, suggesting individual consistency contributes to group parity.

**Orthogonality Rather Than Opposition.** The interaction between fairness and utility metrics (left side of Figure 1) is often negative, confirming the presence of structural tension. However, the magnitude of this conflict is typically small: scores concentrate near zero (median $\mathcal{C} \in [-0.1, 0]$)

rather than approaching $-1$. This suggests that fairness and utility metrics often depend on largely distinct coalition-level subspaces rather than exactly opposing mechanisms. Therefore, compatibility should be interpreted as a structural diagnostic of metric alignment. It helps explain why modest trade-offs can occur in many settings, but it does not by itself determine the shape of the Pareto frontier or guarantee optimization-level independence.

### 4.2. Interaction-Level Attribution

We now investigate the drivers of compatibility by attributing it to specific attribute coalitions.

**Sparsity and Low-Order Dominance.** Figure 2 shows that the vast majority of coalitions have near-zero contribution, confirming the sparsity of metric interactions. Contributions are heavily concentrated in low-order interactions, with the distribution range narrowing significantly as order increases. This aligns with the *conciseness* of interactive concepts (Ren et al., 2023a) and *superior generalization power* of low-order interactions (Ren et al., 2023c; Zhou et al., 2024), suggesting that metric compatibility is primarily determined by simple patterns rather than complex entanglements. Table 6 further reveals a structural asymmetry: supporting coalitions (driving the compatibility direction) are typically **low-order** (e.g., single attributes), while opposing coalitions (resisting the main direction) often involve **higher-order** interactions. This suggests that the dominant compatibility trend (whether synergistic or conflicting) is driven by robust, fundamental features, whereas deviations from this trend typically arise from more complex, noisy patterns.

**The Role of Data Imbalance.** Our fine-grained attribution (Table 6) reveals that the empty set $\emptyset$ often dominates when it appears as a top contributor. In our framework, $\mathbf{I}_m(\emptyset)$ captures the *baseline bias* stemming from dataset-level imbalances. For group fairness metrics under intersectional settings, $\mathbf{I}_m(\emptyset)$ encodes *group imbalance*—when certain sensitive-attribute combinations are underrepresented, we treat them as the minority group (worst-case: zero representation $\Rightarrow$ zero metric value), so $\emptyset$ embeds this structural bias. For utility metrics like accuracy, $\mathbf{I}_{\text{Accuracy}_\downarrow}(\emptyset)$ directly captures the *label imbalance* (base rate $\alpha$). The dominance of $\emptyset$ in pairs like Accuracy-AOD (Table 6: $-476.66\%$) implies that dataset-level biases create a fundamental "baseline effect" that overshadows feature-level interactions. This formalizes observations from prior work (Kamiran & Calders, 2012) and highlights the necessity of data-level interventions (e.g., reweighing, resampling) before model training.

### 4.3. Impact of Debiasing Methods on Compatibility

We examine how debiasing interventions reshape the compatibility landscape. Table 7 reports the average compatibility across metric categories.

**Debiasing Compresses Compatibility Space.** A predominant trend across methods is the reduction in overall average compatibility (Table 7, rightmost column). While Vanilla models achieve $\bar{\mathcal{C}} = 0.2518$, most debiasing methods lower this score, driven by two mechanisms:

1. **Intensified Fairness-Utility Conflict:** The negative compatibility in GF-U generally deepens (e.g., $-0.0380 \rightarrow -0.0507$ for Fair-SMOTE), confirming that enforcing fairness constraints pushes the model away from the optimal utility manifold.

2. **Disruption of Internal Synergy:** Debiasing significantly weakens synergy within fairness families. For instance, GF-IF compatibility drops from 0.41 (Vanilla) to 0.1271 (Fair-SMOTE), suggesting that aggressively targeting one fairness notion (e.g., group parity) often compromises others (e.g., individual consistency).

This trend is not universal—some method-metric combinations show slight compatibility increases (e.g., MAAT's U-U: $0.7454 \rightarrow 0.7823$). We interpret the overall pattern as a *redistribution* of alignment rather than a deterministic law.

**Method-Specific Trade-offs.** The methods exhibit distinct trade-off strategies:

- **Fair-SMOTE (Aggressive Trade-off):** This method yields the lowest overall compatibility (0.1405) and a sharp drop in Utility-Utility synergy ($0.7454 \rightarrow 0.5683$). By aggressively altering the data distribution, it achieves substantial fairness gains but fundamentally disrupts the relationships between all other metrics.

- **MAAT (Gentle Preservation):** In contrast, MAAT maintains the highest compatibility profile (0.2219). Its ensemble-based approach effectively expands the model capacity, allowing it to accommodate fairness constraints without rigidly distorting the decision boundary, thus preserving the natural alignment between utility metrics (U-U increases to 0.7823).

We visualize these structural shifts via network analysis in Appendix B.4.

### 4.4. Case Study: Compatibility Dynamics Under HIFI

We investigate how compatibility evolves along a HIFI debiasing path (Zhang et al., 2025) on Logistic Regression for Census Income ($\eta \in [0, 10^6]$). Figure 3 shows Pareto fronts (left) and compatibility evolution (right). This case study asks whether changes in compatibility can provide a structural explanation for the observed Pareto elbow in this specific setting.

*Table 6.* Top explanatory coalitions for compatibility (Census Income). Top-3 supporting (driving direction) and opposing (resisting) coalitions per pair. Percentages: $\text{Contrib}(C)/\mathcal{C}(m_1, m_2)$ (can exceed 100% when contributions cancel).

| METRIC PAIR | MEAN COMPATIBILITY | TOP-3 SUPPORTING COALITIONS | TOP-3 OPPOSING COALITIONS |
|---|---|---|---|
| ACC vs. AOD | $-0.0125 \pm 0.0372$ | CAPITAL GAIN (113.61%)
EDUCATION-NUM (92.32%)
MARITAL STATUS (59.86%) | $\emptyset$ (-476.66%)
OCCUPATION (-6.04%)
RELATIONSHIP (-4.70%) |
| ACC vs. CFVR | $-0.0236 \pm 0.0111$ | SEX (27.69%)
AGE (25.00%)
AGE, SEX (3.04%) | AGE, MARITAL STATUS, RELATIONSHIP (-2.15%)
AGE, MARITAL STATUS (-1.68%)
AGE, EDUCATION-NUM, RELATIONSHIP (-1.10%) |
| AOD vs. CFVR | $0.5049 \pm 0.0642$ | AGE (27.44%)
SEX (20.54%)
AGE, OCCUPATION (4.51%) | EDUCATION-NUM, RELATIONSHIP, RACE (-0.12%)
EDUCATION-NUM, OCCUPATION, RELATIONSHIP, RACE (-0.11%)
EDUCATION-NUM, MARITAL STATUS, RACE (-0.04%) |

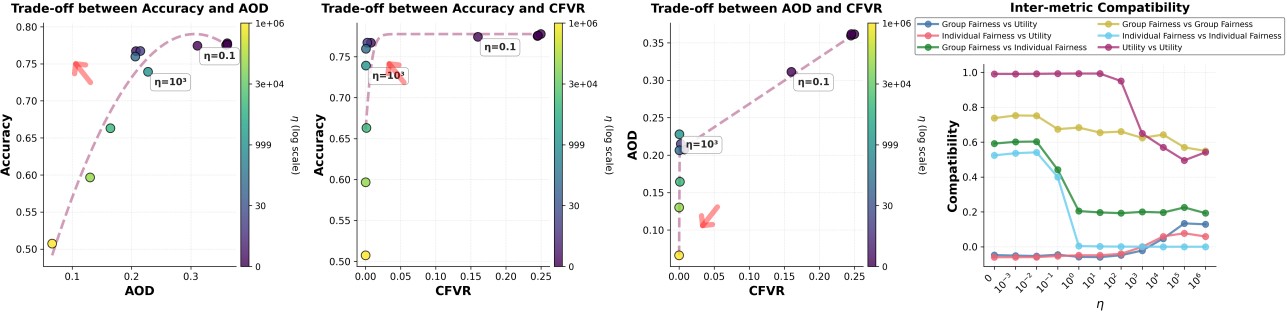

*Figure 3.* Compatibility dynamics in one HIFI case study (LR on Census Income). *Left three panels:* Pareto fronts for three metric pairs as regularization strength $\eta$ increases (color gradient: purple $\rightarrow$ yellow indicates increasing $\eta$). *Rightmost panel:* Evolution of average compatibility $\bar{\mathcal{C}}$ for six categories of metric pairs. *Key finding:* The collapse of Utility-Utility compatibility at $\eta \approx 1000$ (from $\sim 1.0$ to $\sim 0.6$) *coincides precisely* with the "elbow" in all three Pareto fronts, marking the transition from efficient trade-offs to performance degradation. This suggests that compatibility can serve as a diagnostic signal for this regularization path, although we do not claim it is a universal early-warning signal across methods, models, and datasets.

*Table 7.* Average compatibility $\bar{\mathcal{C}}$ for each category of metric pairs under different debiasing methods, aggregated over 6 datasets $\times$ 7 models $\times$ 10 seeds. Abbreviations: Group Fairness (GF), Individual Fairness (IF), Utility (U). *Observation:* All debiasing methods reduce overall compatibility (rightmost column) compared to Vanilla, driven by: (1) intensified Fairness-Utility conflict (GF-U, IF-U become more negative); (2) disrupted internal synergy within fairness families (GF-IF drops sharply).

| METHOD | GF-U | IF-U | GF-IF | GF-GF | IF-IF | U-U | ALL |
|---|---|---|---|---|---|---|---|
| VANILLA | -0.0380 | -0.0340 | 0.4100 | 0.6322 | 0.4154 | 0.7454 | 0.2518 |
| REWEIGHING | -0.0433 | -0.0361 | 0.2882 | 0.4978 | 0.3309 | 0.7499 | 0.1982 |
| FLIPPING | -0.0441 | -0.0381 | 0.2337 | 0.5823 | 0.1692 | 0.7013 | 0.1911 |
| BLINDNESS | -0.0400 | -0.0327 | 0.1454 | 0.5772 | 0.0000 | 0.8005 | 0.1764 |
| FAIR-SMOTE | -0.0507 | -0.0241 | 0.1271 | 0.4715 | 0.2595 | 0.5683 | 0.1405 |
| MAAT | -0.0359 | -0.0352 | 0.3232 | 0.5586 | 0.3448 | 0.7823 | 0.2219 |

**HIFI's Superior Trade-offs and the "Elbow".** HIFI demonstrates effective trade-offs: AOD reduces by $\sim 40\%$ and CFVR approaches zero with negligible accuracy loss for $\eta < 1000$. A sharp "elbow" appears at $\eta \approx 1000$ across all fronts, marking the transition from a "safe" zone (fairness improves "for free") to a "costly" zone (utility degrades).

**Sequential Compression Mechanism.** The rightmost panel of Figure 3 reveals that HIFI compresses compatibility in a specific sequential order as $\eta$ increases:

- **Stage 1: Sacrificing "Cheap" Compatibility** ($\eta \in$ $(0.01, 1]$)**.** Initially, compatibility involving individual fairness (GF-IF and IF-IF) drops sharply, while Utility-Utility (U-U) remains stable at $\sim 1.0$. Individual fairness metrics are loosely coupled with task performance; compressing their consistency yields large fairness gains without disrupting the core utility structure.

- **Stage 2: Utility Structure Collapse** ($\eta > 100$)**.** U-U compatibility suddenly collapses from $\sim 1.0$ to $\sim 0.6$. This collapse *coincides precisely* with the Pareto elbow in all three panels, signaling that the decision boundary is now distorted enough to break the mathematical consistency between utility metrics.

**A Context-Specific Diagnostic Pattern.** In this HIFI+LR+Census setting, the temporal sequence is informative: IF-related compatibility drops first, and U-U compatibility collapses later near the Pareto elbow. This pattern suggests a practical diagnostic heuristic for this case:

- the initial drop in IF-related compatibility corresponds to effective debiasing;

- the later drop in U-U compatibility coincides with utility degradation;

- the useful operating regime lies before U-U compatibility collapses, e.g., $\eta < 1000$ in this experiment.

We emphasize that this observation is currently validated in one case study only. Broader validation across methods, architectures, datasets, and fairness objectives is required before interpreting compatibility collapse as a general early-warning signal.

## 5. Conclusion

We introduced a game-theoretic framework quantifying metric compatibility through interaction-based decomposition, enabling mechanistic analysis without causal graphs. Our study (6 datasets, 7 models, 6 methods) reveals: (1) fairness-utility metrics exhibit orthogonality (median $\mathcal{C} \approx 0$) driven by sparse, low-order interactions; (2) debiasing compresses compatibility space (Fair-SMOTE: 44.2% vs. MAAT: 11.87%), redistributing rather than eliminating conflicts; (3) compatibility is context-specific, necessitating empirical evaluation per application. Our framework can enable informed metric selection, targeted interventions via coalition attribution, and context-specific compatibility monitoring during debiasing. Future work should extend to high-dimensional/unstructured data and formalize compatibility-Pareto relationships.

## Acknowledgements

We thank the anonymous reviewers for their constructive comments and helpful suggestions.

## Impact Statement

This paper presents work whose goal is to advance the field of machine learning fairness through better understanding of metric relationships. We acknowledge several potential broader impacts:

**Positive Impacts.** Our framework provides mechanistic explanations of fairness-utility trade-offs without requiring causal graphs or unbiased labels, potentially lowering barriers to fairness analysis for practitioners without causal inference expertise. By revealing that fairness and utility metrics are often orthogonal rather than sharply conflicting, our work may encourage broader adoption of fairness interventions by dispelling the myth of prohibitive costs. The coalition-level attribution enables targeted debugging of algorithmic bias (e.g., identifying which features drive conflicts).

**Limitations and Risks.** Our framework explains *why* trade-offs exist but does not prescribe *which* fairness metric should be prioritized—a fundamentally normative decision requiring stakeholder input and domain expertise. Our finding that

compatibility is highly context-dependent means practitioners cannot bypass empirical evaluation for their specific application. Importantly, achieving metric parity does not guarantee substantive justice or eliminate disparate impact in unmeasured dimensions; our tools should complement—not replace—participatory approaches to algorithmic accountability (e.g., community audits, impact assessments).

**Dual-Use Considerations.** Our decomposition methods could theoretically be misused to identify minimal interventions that satisfy fairness metrics while preserving discriminatory patterns in unmeasured dimensions ("fairness washing"). We urge practitioners to use our framework as a *diagnostic tool within broader fairness audits* rather than as a compliance checklist. Specifically:

- Compatibility analysis should inform—not replace—Pareto analysis and stakeholder deliberation.

- High compatibility between a fairness metric and utility does not imply the metric is "easy to satisfy" or less important.

- Practitioners should validate that debiasing interventions improve real-world outcomes, not just metric values.

**Broader Context.** Our work is part of a growing effort to make fairness interventions more transparent and interpretable. We hope the compatibility framework will facilitate more nuanced conversations about trade-offs in fairness-aware ML, moving beyond binary narratives of "fairness vs. accuracy" toward a richer understanding of metric geometry.

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

# A. Additional Details

## A.1. GIFVR Definition and Relationship to CFVR

For the Global Individual Fairness Violation Ratio (GIFVR) metric, we define a pair of samples as similar if their non-sensitive attributes exhibit small differences, *regardless of differences in their sensitive attributes*. This captures the intuition that individuals with similar qualifications (non-sensitive features) should receive similar predictions, even if they belong to different demographic groups.

**Formal Definition.** For the definition of GIFVR in Table 1, $d : \mathcal{D}_N \times \mathcal{D}_N \to \{0, 1\}$ is a similarity metric that evaluates to 1 when $x$ and $x'$ are similar, where $\mathcal{D}_N$ is the input domain. Let $x = (ns, s)$ and $x' = (ns', s')$, where $ns, ns' \in NS$ are non-sensitive attribute vectors, and $s, s' \in S$ are sensitive attributes. For each dimension $i$ of the non-sensitive attributes, let $r_i = u_i - l_i$ denote the range, where $[l_i, u_i]$ is the domain of the $i$-th non-sensitive attribute. Formally:

$$
d(x, x') = \begin{cases} 1 & \text{if } s, s' \in S \text{ and } \forall i \in \{1, \ldots, |N \setminus A|\} : \\ & \quad |ns_i - ns'_i| = 0 \text{ if } r_i \cdot c < 1 \\ & \quad |ns_i - ns'_i| \le 1 \text{ if } r_i \cdot c \ge 1 \\ 0 & \text{otherwise} \end{cases}
\tag{5}
$$

where $c \in \mathbb{R}_+$ is a threshold parameter controlling the strictness of similarity. In our experiments, we set $c = 0.2$. Note that for discretized features, 1 represents the minimum non-zero difference.

**Relationship to CFVR.** Causal Fairness Violation Ratio (CFVR) is a special case of GIFVR with $c = 0$, meaning only samples with *identical* non-sensitive attributes are considered similar. This stricter definition focuses on causal fairness: "Would the prediction change if only the sensitive attribute were different?" In contrast, GIFVR with $c > 0$ allows for approximate similarity, capturing a broader notion of individual fairness.

## A.2. Metric Decomposition

Here, we present the decomposition derivations for each approximate metric:

$$
\begin{aligned}
\widetilde{\text{SPD}} &\triangleq \mathbb{E}_{x \sim P_p} v(x) - \mathbb{E}_{x \sim P_{up}} v(x) \\
&= \mathbb{E}_{x \sim P_p} \sum_{C \subseteq N} I_C(x) - \mathbb{E}_{x \sim P_{up}} \sum_{C \subseteq N} I_C(x) \\
&= \sum_{C \subseteq N} \left[ \mathbb{E}_{x \sim P_p} I_C(x) - \mathbb{E}_{x \sim P_{up}} I_C(x) \right]
\end{aligned}
\tag{6}
$$

$$
\begin{aligned}
\widetilde{\text{EOD}} &\triangleq \mathbb{E}_{x \sim P_{p \& f}} v(x) - \mathbb{E}_{x \sim P_{up \& f}} v(x) \\
&= \mathbb{E}_{x \sim P_{p \& f}} \sum_{C \subseteq N} I_C(x) - \mathbb{E}_{x \sim P_{up \& f}} \sum_{C \subseteq N} I_C(x) \\
&= \sum_{C \subseteq N} \left[ \mathbb{E}_{x \sim P_{p \& f}} I_C(x) - \mathbb{E}_{x \sim P_{up \& f}} I_C(x) \right]
\end{aligned}
\tag{7}
$$

$$
\begin{aligned}
\widetilde{\text{PED}} &\triangleq \mathbb{E}_{x \sim P_{p \& uf}} v(x) - \mathbb{E}_{x \sim P_{up \& uf}} v(x) \\
&= \mathbb{E}_{x \sim P_{p \& uf}} \sum_{C \subseteq N} I_C(x) - \mathbb{E}_{x \sim P_{up \& uf}} \sum_{C \subseteq N} I_C(x) \\
&= \sum_{C \subseteq N} \left[ \mathbb{E}_{x \sim P_{p \& uf}} I_C(x) - \mathbb{E}_{x \sim P_{up \& uf}} I_C(x) \right]
\end{aligned}
\tag{8}
$$

$$\widetilde{\text{AOD}} \triangleq \frac{1}{2} \left( \left( \mathbb{E}_{x \sim P_{p\&f}} v(x) + \mathbb{E}_{x \sim P_{p\&uf}} v(x) \right) \right.$$
$$\left. - \left( \mathbb{E}_{x \sim P_{up\&f}} v(x) + \mathbb{E}_{x \sim P_{up\&uf}} v(x) \right) \right)$$
$$= \frac{1}{2} \left[ \left( \mathbb{E}_{x \sim P_{p\&f}} \sum_{C \subseteq N} I_C(x) + \mathbb{E}_{x \sim P_{p\&uf}} \sum_{C \subseteq N} I_C(x) \right) \right.$$
$$\left. - \left( \mathbb{E}_{x \sim P_{up\&f}} \sum_{C \subseteq N} I_C(x) + \mathbb{E}_{x \sim P_{up\&uf}} \sum_{C \subseteq N} I_C(x) \right) \right] \tag{9}$$
$$= \frac{1}{2} \sum_{C \subseteq N} \left[ \left( \mathbb{E}_{x \sim P_{p\&f}} I_C(x) + \mathbb{E}_{x \sim P_{p\&uf}} I_C(x) \right) \right.$$
$$\left. - \left( \mathbb{E}_{x \sim P_{up\&f}} I_C(x) + \mathbb{E}_{x \sim P_{up\&uf}} I_C(x) \right) \right]$$

$$\widetilde{\text{GIFVR}} \triangleq \mathbb{E}_x \left[ \max_{x':d(x,x')=1} v(x') - \min_{x':d(x,x')=1} v(x') \right]$$
$$= \mathbb{E}_x \left[ v(x^+) - v(x^-) \right]$$
$$= \mathbb{E}_x \left[ \sum_{C \subseteq N} I_C(x^+) - \sum_{C \subseteq N} I_C(x^-) \right] \tag{10}$$
$$= \sum_{C \subseteq N} \mathbb{E}_x \left[ I_C(x^+) - I_C(x^-) \right]$$
$$(x^+ = argmax_{x':d(x,x')=1} v(x'),$$
$$x^- = argmin_{x':d(x,x')=1} v(x'))$$

$$\widetilde{\text{CFVR}} \triangleq \mathbb{E}_{ns} \max_{s,s' \in S} [v(ns,s) - v(ns,s')]$$
$$= \mathbb{E}_{ns} [v(ns,s_p) - v(ns,s_{up})]$$
$$= \mathbb{E}_{ns} \left[ \sum_{C \subseteq N} I_C(ns,s_p) - \sum_{C \subseteq N} I_C(ns,s_{up}) \right]$$
$$= \sum_{C \subseteq N} \mathbb{E}_{ns} [I_C(ns,s_p) - I_C(ns,s_{up})] \tag{11}$$
$$= \sum_{C \subseteq N \wedge C \cap A \neq \emptyset} \mathbb{E}_{ns} [I_C(ns,s_p) - I_C(ns,s_{up})] \text{(Assume using global baseline values.)}$$
$$(s_p = argmax_{s \in S} v(ns,s),$$
$$s_{up} = argmin_{s \in S} v(ns,s))$$

$$\widetilde{\text{Accuracy}_\downarrow} \triangleq 1 - \left[ \alpha \cdot \mathbb{E}_{x \sim P_f} v(x) + (1-\alpha) \cdot \mathbb{E}_{x \sim P_{uf}} [1 - v(x)] \right]$$
$$= 1 - \left[ \alpha \cdot \mathbb{E}_{x \sim P_f} \sum_{C \subseteq N} I_C(x) - (1-\alpha) \cdot \mathbb{E}_{x \sim P_{uf}} \sum_{C \subseteq N} I_C(x) + 1 - \alpha \right] \tag{12}$$
$$= \sum_{C \subseteq N} \left[ (1-\alpha) \cdot \mathbb{E}_{x \sim P_{uf}} I_C(x) - \alpha \cdot \mathbb{E}_{x \sim P_f} I_C(x) \right] + \alpha$$
$$(\alpha = P[Y=1], \text{ and } \mathbf{I}_{\text{Accuracy}_\downarrow}(\emptyset) \text{ includes the constant } \alpha.)$$

$$
\begin{aligned}
\widetilde{\mathrm{Recall}_\downarrow} &\triangleq 1 - \mathbb{E}_{x \sim P_f} v(x) \\
&= 1 - \mathbb{E}_{x \sim P_f} \sum_{C \subseteq N} I_C(x) \\
&= 1 - \sum_{C \subseteq N} \mathbb{E}_{x \sim P_f} I_C(x)
\end{aligned}
\tag{13}
$$

($\mathbf{I}_{\mathrm{Recall}_\downarrow}(\emptyset)$ includes the constant 1.)

$$
\begin{aligned}
\widetilde{\mathrm{FPR}} &\triangleq \mathbb{E}_{x \sim P_{uf}} v(x) \\
&= \mathbb{E}_{x \sim P_{uf}} \sum_{C \subseteq N} I_C(x) \\
&= \sum_{C \subseteq N} \mathbb{E}_{x \sim P_{uf}} I_C(x)
\end{aligned}
\tag{14}
$$

### A.3. Proof of Proposition 3.2

We provide the detailed derivation for Proposition 3.2. Recall the definition of $\widetilde{\mathrm{CFVR}}$:

$$
\widetilde{\mathrm{CFVR}} \triangleq \mathbb{E}_{ns} \left[ \max_{s,s' \in S} (v(ns, s) - v(ns, s')) \right].
\tag{15}
$$

Let $s_p$ and $s_{up}$ denote the sensitive attribute values that maximize and minimize the output for a given $ns$, respectively. Substituting the Harsanyi decomposition $v(x) = \sum_{C \subseteq N} I_C(x)$, we get:

$$
\begin{aligned}
\widetilde{\mathrm{CFVR}} &= \mathbb{E}_{ns}[v(ns, s_p) - v(ns, s_{up})] \\
&= \mathbb{E}_{ns} \left[ \sum_{C \subseteq N} I_C(ns, s_p) - \sum_{C \subseteq N} I_C(ns, s_{up}) \right] \\
&= \sum_{C \subseteq N} \mathbb{E}_{ns}[I_C(ns, s_p) - I_C(ns, s_{up})].
\end{aligned}
\tag{16}
$$

Now, consider a coalition $C$ that consists solely of non-sensitive attributes (i.e., $C \subseteq N \setminus A$, or $C \cap A = \emptyset$). The Harsanyi interaction $I_C(x)$ is calculated using $v(x_L)$ terms where $L \subseteq C$. Since $L$ contains only non-sensitive attributes, the sensitive attributes are always in the masked set $N \setminus L$. **Under the assumption of global baseline values** (where masked features are replaced by a fixed constant $b$), the sensitive attributes are replaced by the same baseline values $b_A$ regardless of whether the original input was $s_p$ or $s_{up}$. Consequently, $v((ns, s_p)_L) = v((ns, s_{up})_L)$, which implies $I_C(ns, s_p) = I_C(ns, s_{up})$.

Therefore, the term inside the expectation vanishes for all $C \subseteq N \setminus A$:

$$
\mathbb{E}_{ns}[I_C(ns, s_p) - I_C(ns, s_{up})] = 0, \quad \forall C \subseteq N \setminus A.
\tag{17}
$$

This simplifies the summation to only include coalitions involving sensitive attributes:

$$
\widetilde{\mathrm{CFVR}} = \sum_{C \subseteq N, C \cap A \neq \emptyset} \mathbb{E}_{ns}[I_C(ns, s_p) - I_C(ns, s_{up})].
\tag{18}
$$

This concludes the proof.

## B. Additional Empirical Results

This appendix provides comprehensive details and extended results for the experiments reported in Sec. 4. We organize the content as follows:

- Appendix B.1: Dataset characteristics and preprocessing details

- Appendix B.2: Evaluation of debiasing methods across all metrics

- Appendix B.3: PCA visualization of metric clustering in representation space

- Appendix B.4: Network visualization of compatibility structure

- Appendix B.5: Training dynamics of compatibility and attribute importance

- Appendix B.6: Factor analysis (ANOVA) and predictability experiments

- Appendix B.7: Coalition attribution results for all datasets

- Appendix B.8: Robustness to fixed baseline choices

- Appendix B.9: Distributional masking with marginal sampling

**Experimental Protocol.** Unless otherwise specified, all experiments follow this protocol:

1. **Data Splitting:** 70% for training and 30% for test.

2. **Interaction Computation:** For computational efficiency, we sample 500 test instances (or use the full test set if $|\text{test}| < 500$) to compute Harsanyi interactions. This sampling is justified by the strong correlations in Table 4.

3. **Random Seeds:** All configurations are repeated with 10 random seeds controlling data splits, model initialization, and stochastic training procedures.

4. **Baseline Values:** For Harsanyi interaction computation, we use the *mean* of each feature in the training set as the baseline (masked value). Although the mean baseline is efficient and uniform across datasets, it can create out-of-distribution masked inputs for encoded categorical variables. We therefore evaluate median, mode, and marginal-sampling alternatives in Appendix B.8 and Appendix B.9.

### B.1. Datasets

We provide the details of the experimental datasets in Table 8. To ensure reproducibility, we provide detailed descriptions of our data preprocessing pipeline for all six tabular datasets used in our experiments.

Our preprocessing follows a consistent four-stage pipeline across all datasets:

**Stage 1: Data Cleaning.** We remove samples with missing values (N/A) and drop redundant or irrelevant attributes. For datasets with categorical variables containing "unknown" values, we impute them with the mode of the corresponding feature.

**Stage 2: Feature Encoding.** We apply the following encoding strategies:

- **Sensitive attributes**: Binary encoding based on domain-specific thresholds (e.g., age $\geq 40$ for Census, age $< 50$ for Heart Disease).

- **Categorical features**: Converted to numerical codes using category encoding.

- **Continuous features**: Discretized into bins using either uniform binning (e.g., 10 bins for balance in Bank) or domain-specific bins (e.g., glucose levels in Diabetes).

**Stage 3: Label Processing.** Target labels are binarized to ensure consistency, where 1 typically represents the positive outcome (e.g., income $>$50K for Census).

**Stage 4: Data Splitting and Normalization.** For each random seed in $\{0, 1, 2, 3, 4, 5, 6, 7, 8, 9\}$:

- Split data into training and test sets.

*Table 8.* Datasets. We report the statistics for each dataset including the number of samples (# Samp.), features (# Feat.), and protected attributes (Prot. Attr.). We also report: Imbal., the minority-to-majority class ratio; Avg. Corr., the mean absolute Cramér's V between non-sensitive attributes and the intersectional sensitive attribute (aggregated by Cartesian product, e.g., "Young-Black-Female"); and Mut. Inf., the normalized mutual information between the intersectional sensitive attribute and the label. Lower Imbal. means more severe imbalance; higher Avg. Corr. suggests stronger feature-group entanglement; higher Mut. Inf. indicates more severe label bias.

| NAME | TASK | # SAMP. | # FEAT. | PROT. ATTR. | IMBAL. | AVG. CORR. | MUT. INF. | REF. |
|---|---|---|---|---|---|---|---|---|
| CENSUS INCOME | INCOME PREDICTION | 30162 | 12 | AGE, RACE, SEX | 0.3290 | 0.1573 | 0.0468 | (BECKER & KOHAVI, 1996) |
| UFRGS | GPA PREDICTION | 43303 | 6 | SEX, AGE, RACE | 0.8774 | 0.1379 | 0.0250 | (DA SILVA, 2019) |
| COMPAS | RE-OFFENCE PREDICTION | 7214 | 11 | SEX, RACE | 0.8246 | 0.1368 | 0.0149 | (PROPUBLICA, 2016) |
| DIABETES | READMISSION PREDICTION | 768 | 8 | AGE | 0.5349 | 0.2915 | 0.0675 | (KAHN, 1994) |
| BANK MARKETING | TERM DEPOSIT PREDICTION | 45211 | 12 | AGE | 0.1328 | 0.1057 | 0.0002 | (MORO ET AL., 2014) |
| HEART DISEASE | HEART DISEASE PREDICTION | 296 | 13 | AGE, SEX | 0.8459 | 0.2149 | 0.0784 | (JÁNOSI ET AL., 1989) |

- Fit a StandardScaler on training features and save for consistent normalization.

- Compute and save baseline values (feature-wise means) and constraints (min/max ranges) from training data.

- Sample a fixed number of test instances (default: 500) for efficient analysis.

Other data-specific preprocessing details are listed below:

**Census Income.** We drop the redundant "Education" and "fnlwgt" attributes. Age is binarized at threshold 40, race is mapped to White vs. non-White, and sex to Male vs. Female. Capital gain/loss are scaled by division (10000 and 500 respectively), and hours per week by 10.

**UFRGS.** Exam scores (physics, biology, etc.) are discretized into 8 bins based on score ranges [0, 300, ..., 900, 2000]. Gender is encoded as Male (1) vs. Female (0), and race as White (1) vs. non-White (0).

**COMPAS.** We retain only core features and drop 38 irrelevant attributes. Age is discretized into three groups: $< 25$ (0), 25-45 (1), and $> 45$ (2). Race is binarized to Caucasian vs. non-Caucasian, and priors count is discretized into 10 bins.

**Diabetes.** Age is binarized at threshold 30. Medical measurements are discretized: glucose (9 bins), blood pressure (9 bins), skin thickness (10 bins), insulin (11 bins), BMI (6 bins), and diabetes pedigree function (13 bins).

**Bank Marketing.** We drop temporal features (contact, day, month, duration) and impute missing categorical values with mode. Numerical features (balance, campaign, pdays, previous) are discretized into 10 bins each. Age is binarized at threshold 40.

**Heart Disease.** We filter out erroneous records (ca $\geq 4$ or thal $= 0$). Four continuous features (resting blood pressure, cholesterol, max heart rate, ST depression) are discretized into 10 bins each. Age is binarized at threshold 50.

Our current experiments do not include a systematic ablation over bin granularity. We therefore do not claim invariance to arbitrary discretization schemes. Instead, our preprocessing uses domain thresholds where available and balanced or range-based binning for continuous variables. The consistency of the main compatibility patterns across six datasets with different feature types and discretization characteristics suggests robustness to reasonable discretization choices, but a full granularity ablation remains future work.

## B.2. Evaluation of Debiasing Methods

We have comprehensively evaluated the vanilla model, Reweighing (Kamiran & Calders, 2012), flipping-based retraining (Li et al., 2023), fairness through blindness (Dwork et al., 2012; Yang et al., 2024), Fair-SMOTE (Chakraborty et al., 2021), and MAAT (Chen et al., 2022) on all metrics in Table 1, 2 and 3. The evaluation results in Table 9 are averaged across 6 datasets, 7 ML models, and 10 random seeds.

## B.3. Metric Clustering in Representation Space

To visualize the global relationships between metric vectors, we apply Principal Component Analysis (PCA) to the interaction vectors $\mathbf{I}_m$ derived from all 420 experiments on vanilla models. Figure 4 presents the 2D projection for each dataset.

**Fairness Clustering vs. Utility Dispersion.** A consistent pattern across all datasets is the relatively tight clustering of fairness metrics (both group and individual). This suggests that despite their differing mathematical formulations, they share

*Table 9.* Comparison of fairness and utility metrics across debiasing methods, averaged over 6 datasets × 7 models × 10 seeds (420 configurations). **Original** rows show metrics computed on discrete predictions using the full test set; **Approximate** rows show continuous proxies (Table 3) computed on 500 sampled test instances. For each original metric, the best value among the five debiasing methods is **bolded**, and the worst is underlined. *Observation:* All debiasing methods reduce fairness violations compared to Vanilla, but with varying utility costs. Fair-SMOTE achieves the largest group fairness improvements but also the largest accuracy drop (0.7692 → 0.7207). MAAT achieves a favorable balance, improving fairness with minimal utility loss.

| METHOD | TYPE | SPD↓ | EOD↓ | PED↓ | AOD↓ | GIFVR↓ | CFVR↓ | ACCURACY↑ | PRECISION↑ | RECALL↑ | FPR↓ | F1-SCORE↑ | MCC↑ | ROC-AUC↑ |
|---|---|---|---|---|---|---|---|---|---|---|---|---|---|---|
| VANILLA | ORIGINAL | 0.4183 | 0.3403 | 0.3228 | 0.3483 | 0.5190 | 0.2625 | 0.7692 | 0.7390 | 0.6911 | 0.2226 | 0.6990 | 0.4187 | 0.7822 |
| VANILLA | APPROXIMATE | 0.3016 | 0.2618 | 0.2330 | 0.2641 | 0.4136 | 0.1871 | 0.6934 | / | 0.5636 | 0.3020 | / | / | / |
| REWEIGHING | ORIGINAL | 0.3445 | 0.2796 | 0.2535 | 0.2752 | 0.4980 | 0.2174 | 0.7651 | 0.7346 | 0.6861 | 0.2300 | 0.6923 | 0.4101 | 0.7786 |
| REWEIGHING | APPROXIMATE | 0.2648 | 0.2321 | 0.2086 | 0.2291 | 0.4023 | 0.1634 | 0.6892 | / | 0.5641 | 0.3089 | / | / | / |
| FLIPPING | ORIGINAL | 0.3364 | 0.2593 | 0.2411 | 0.2610 | 0.4796 | 0.1325 | 0.7650 | 0.7340 | 0.6865 | 0.2274 | **0.6943** | 0.4097 | 0.7768 |
| FLIPPING | APPROXIMATE | 0.2516 | 0.2202 | 0.1899 | 0.2179 | 0.3925 | 0.0890 | 0.6933 | / | 0.5658 | 0.3030 | / | / | / |
| BLINDNESS | ORIGINAL | 0.2699 | 0.2030 | **0.1846** | 0.1975 | 0.4139 | **0.0000** | 0.7625 | 0.7330 | 0.6818 | 0.2290 | 0.6900 | 0.4034 | 0.7734 |
| BLINDNESS | APPROXIMATE | 0.2082 | 0.1960 | 0.1545 | 0.1864 | 0.3327 | 0.0000 | 0.6864 | / | 0.5564 | 0.3092 | / | / | / |
| FAIR-SMOTE | ORIGINAL | **0.2616** | **0.1980** | 0.1910 | **0.1956** | 0.5402 | 0.1400 | 0.7207 | 0.6817 | **0.6929** | 0.2614 | 0.6776 | 0.3732 | 0.7533 |
| FAIR-SMOTE | APPROXIMATE | 0.2411 | 0.2434 | 0.2088 | 0.2206 | 0.4959 | 0.1493 | 0.6611 | / | 0.6110 | 0.3226 | / | / | / |
| MAAT | ORIGINAL | 0.3400 | 0.2739 | 0.2420 | 0.2686 | 0.4878 | 0.2063 | **0.7687** | **0.7443** | 0.6811 | **0.2195** | 0.6909 | **0.4120** | **0.7854** |
| MAAT | APPROXIMATE | 0.2496 | 0.2237 | 0.1823 | 0.2145 | 0.3816 | 0.1458 | 0.6918 | / | 0.5444 | 0.2980 | / | / | / |

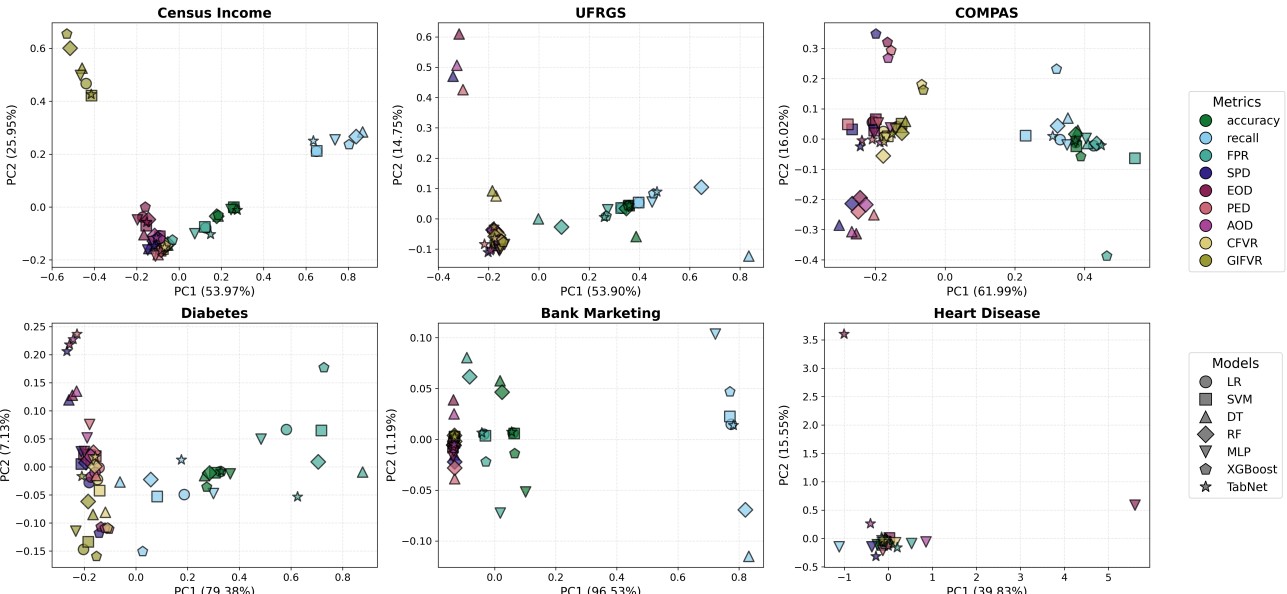

*Figure 4.* PCA visualization of metric interaction vectors across six datasets. Each point represents a metric vector $\mathbf{I}_m$ for a specific model instance. Colors denote metric types, and shapes denote model architectures. Fairness metrics consistently cluster together, while utility metrics disperse, illustrating their distinct structural dependencies.

a common underlying representation structure. In contrast, utility metrics (Accuracy, Recall, FPR) tend to disperse along the first principal component (PC1), indicating that they rely on distinct feature mechanisms.

**Dominance of Data over Model.** The clustering patterns are largely consistent across different model architectures (represented by different shapes in Figure 4). While some model-specific effects are visible (e.g., MLP and TabNet show distinct outliers for AOD on the Heart Disease dataset, likely due to overfitting on small data), the overall topology is dictated by the dataset. This suggests that the intrinsic compatibility between metrics is primarily a property of the *data distribution* and the *metric definitions* themselves, rather than an artifact of specific model choices. This finding is further corroborated by our ANOVA results (Figure 7), where the Dataset factor and Dataset × Model interaction dominate compatibility variance. This conclusion is based on PCA visualization (dimensionality reduction), which may obscure some model-specific patterns in high-dimensional space. However, the consistency with ANOVA results (which analyze the full $2^{|N|}$-dimensional vectors) increases our confidence in this interpretation.

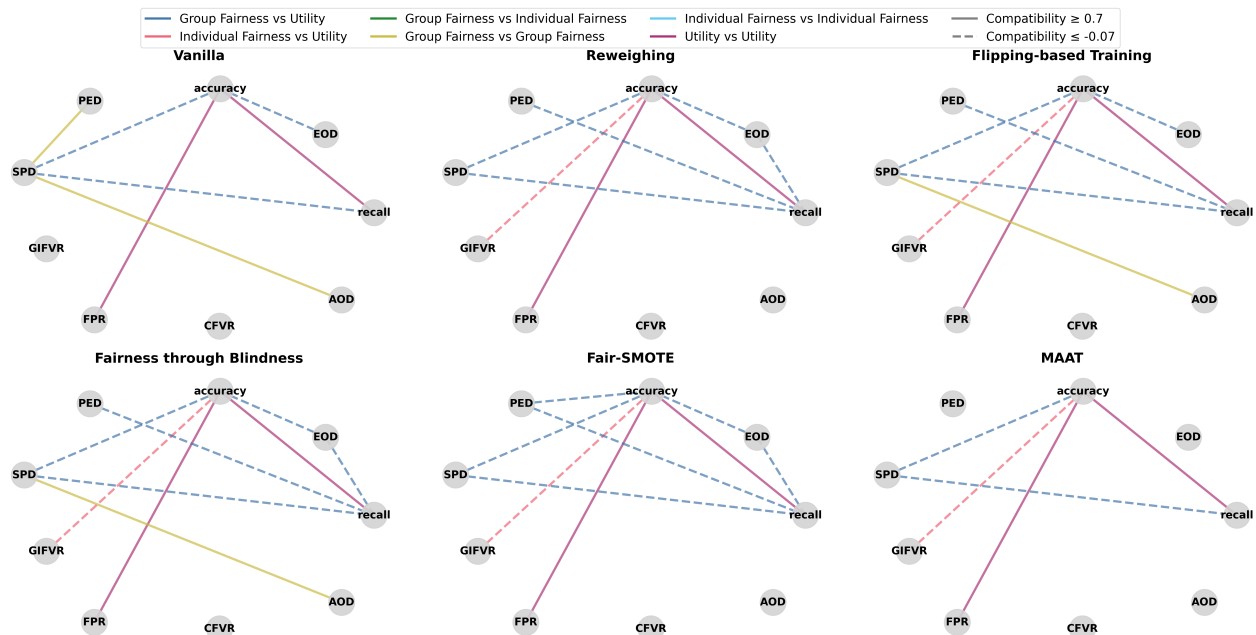

*Figure 5.* Compatibility networks across debiasing methods. Nodes represent metrics, and edges denote strong interactions (Solid: $\mathcal{C} \geq 0.7$, Dashed: $\mathcal{C} \leq -0.07$). Compared to Vanilla, debiasing methods (especially Fair-SMOTE) reduce the number of synergistic links (solid) and increase the number of conflicting links (dashed), visually demonstrating the trade-off mechanism.

## B.4. Visualizing Debiasing Effects on Metric Compatibility

To visualize the structural changes induced by debiasing, we construct compatibility networks for each method (Figure 5). Nodes represent metrics, and edges represent strong interactions: solid lines for high compatibility ($\mathcal{C} \geq 0.7$) and dashed lines for notable conflict ($\mathcal{C} \leq -0.07$).

The visualization confirms the "compression" effect discussed in Section 4.3:

- **Loss of High-Compatibility Edges:** The Vanilla network features 4 solid edges, connecting both utility metrics (e.g., Accuracy-Recall) and group fairness metrics (e.g., AOD-SPD). Debiasing methods typically remove the fairness-fairness links. For instance, Reweighing and Fair-SMOTE lose the strong connections within the group fairness cluster, leaving mostly utility-utility edges intact.

- **Proliferation of Conflict Edges:** The number of dashed conflict lines increases significantly with debiasing. While Vanilla exhibits 3 conflict edges (GF-U), methods like Fair-SMOTE and Blindness display up to 6-7 conflict edges, extending the tension to include Individual-Utility pairs (e.g., Accuracy-GIFVR).

This network topology illustrates that debiasing does not simply "fix" fairness; it reconfigures the entire metric ecosystem, often replacing a landscape of synergy with one of heightened tension.

## B.5. How Does Compatibility Evolve During Training?

To understand the temporal dynamics of metric interactions, we track the evolution of compatibility and attribute importance across 10 checkpoints during the training of a TabNet model on the Census Income dataset. Figure 6 visualizes these dynamics, with the left panel showing the compatibility evolution for all 36 metric pairs and the right panel displaying the changing relative importance of input attributes.

**Taxonomy of Compatibility Evolution Patterns.** Based on the trajectories observed in the left panel, we categorize the evolution into seven distinct patterns, largely determined by the types of metrics involved:

- **Utility-Driven Volatility (Types A, B, C):** Pairs involving utility metrics often exhibit initial instability due to the rapid changes in the decision boundary during early learning.

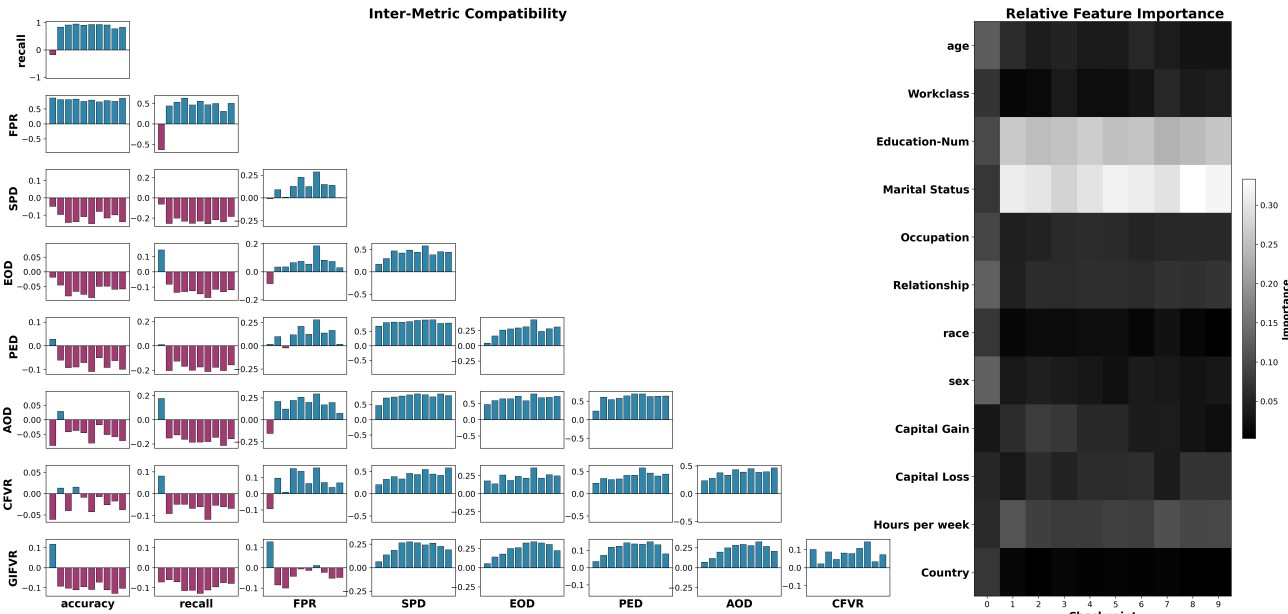

*Figure 6.* Training dynamics of metric compatibility and attribute importance. *Left:* Evolution of compatibility scores for 36 metric pairs across 10 training checkpoints. The bars in each cell represent the compatibility value at each checkpoint, revealing distinct patterns of stability, oscillation, and reversal. *Right:* Heatmap of relative attribute importance over training time. Brighter colors indicate higher importance. The model progressively concentrates on a few dominant features (e.g., Marital Status) while suppressing sensitive attributes (age, race, sex).

- *Type A (Rapid Correction & Oscillation):* 7 pairs (e.g., Accuracy-Recall, Recall-FPR) start with a direction opposite to the final state but rapidly correct and then oscillate. This reflects the initial phase where the model has not yet learned task-relevant features.
- *Type B (Correction-Strengthen-Weaken):* 5 pairs (e.g., Recall-EOD, FPR-SPD) correct their initial direction, strengthen the alignment as the model learns, but then weaken slightly, possibly due to overfitting in later stages.
- *Type C (Mid-Training Reversal):* 4 pairs (e.g., Accuracy-AOD, FPR-GIFVR) show a temporary reversal in compatibility direction during the middle stages, highlighting a transient conflict between task learning and fairness objectives.

- **Structural Stability (Type D):** Pairs with highly correlated mathematical definitions, such as Accuracy-FPR and SPD-PED, remain directionally consistent throughout training with minimal fluctuation.

- **Fairness Synergy Evolution (Types E, F, G):** Pairs involving fairness metrics generally show more stable, positive trends.

  - *Type E (Group Fairness):* 7 pairs involving group fairness (e.g., Accuracy-SPD, SPD-EOD) strengthen their alignment early and then stabilize, suggesting group-level parities are established relatively quickly.
  - *Type F (Individual Fairness):* 9 pairs, mostly involving individual fairness (e.g., AOD-GIFVR, CFVR-GIFVR), strengthen initially but weaken in later stages. This implies that as the model overfits to specific samples to minimize loss, it may compromise the robustness required for individual fairness.
  - *Type G (Group-CFVR Synergy):* Uniquely, SPD-CFVR and AOD-CFVR show a continuous upward trend, indicating that the synergy between group parity and causal fairness grows stronger as the model matures.

**Dynamics of Attribute Importance.** The right panel of Figure 6 reveals a clear shift in feature reliance during training. The model rapidly focuses on task-critical attributes: "Marital Status", "Education", and "Hours per week" see their importance surge, eventually accounting for $\sim 70\%$ of the total importance. Conversely, sensitive attributes ("age", "race", "sex") are suppressed, with their importance diminishing to negligible levels.

The attribute importance shown in Figure 6 (right panel) is computed using TabNet's built-in attention mechanism (Arik & Pfister, 2021), which aggregates the attention weights across all decision steps during forward propagation. This is distinct

*Figure 7.* Factor analysis of compatibility variance (ANOVA). The bar charts show the effect size ($\omega^2$) of Dataset, Model, Method, and their interactions on the compatibility of three representative metric pairs. For Fairness-Utility pairs (left/middle), the **Dataset** $\times$ **Model** interaction dominates, indicating complex dependencies. For the Fairness-Fairness pair (right), the **Method** choice is the primary driver.

from Shapley values or other post-hoc attribution methods. TabNet's attention provides a model-intrinsic measure of which features the model actively uses for prediction.

This suppression creates an inherent tension in fairness optimization. While the model naturally learns to reduce the *attention* on sensitive attributes to maximize utility (thereby reducing *explicit* reliance on protected features), many fairness metrics implicitly require the model to account for these attributes to adjust predictions for different groups. The "natural" learning path of the model thus diverges from the requirements of certain fairness definitions, explaining why post-hoc debiasing often requires aggressive interventions.

**Attention Suppression vs. Interaction-Level Influence.** Crucially, the suppression of attention weights (TabNet's importance) does not imply that sensitive attributes are irrelevant for compatibility. As shown in Tables 6, 10, 11, and 12, coalitions containing sensitive attributes often have the highest explanatory power for compatibility. This apparent paradox is resolved by distinguishing between two distinct concepts:

- **TabNet Attention (Model-Intrinsic Importance):** TabNet's attention mechanism measures which features the model *actively queries* during prediction. Low attention on "sex" means the model does not directly use this feature in its decision-making process (at least not through the attention pathway).

- **Harsanyi Interaction (Compatibility Driver):** In contrast, $\mathbf{I}_m(C)$ for a specific coalition $C$ (e.g., $C = \{\text{sex}\}$ or $C = \{\text{sex}, \text{marital status}\}$) measures how that particular attribute pattern *contributes to the metric value*. Even when TabNet assigns low attention to "sex", the *coalitions* involving "sex" can still have large $|\mathbf{I}_m(C)|$, driving compatibility or conflict between metrics.

This distinction highlights why interaction-based decomposition is necessary for understanding metric relationships:

- **Model-intrinsic importance** (e.g., TabNet attention, gradient-based saliency) reveals what the model *thinks* is important for its predictions, but may not capture the patterns that drive fairness violations.

- **Coalition-level attribution** (our framework) reveals *which specific attribute combinations* drive compatibility or conflict between fairness and utility metrics, enabling targeted interventions. For example, even if a model suppresses attention on "sex", we can identify that conflicts arise from the $\{\text{sex}, \text{marital status}\}$ interaction pattern in the data distribution.

In summary, TabNet suppresses *attention* on sensitive attributes to avoid explicit bias, but the *interactive patterns* involving these attributes (captured by Harsanyi interactions) remain the primary drivers of fairness-utility compatibility.

### B.6. Can We Predict Compatibility?

To understand the sources of compatibility variance, we perform a Type II ANOVA with interaction terms on three representative metric pairs: Accuracy-AOD (Group Fairness vs. Utility), Accuracy-CFVR (Individual Fairness vs. Utility), and AOD-CFVR (Group Fairness vs. Individual Fairness). The factors are: **Dataset** (6 levels), **Model** (7 levels), and **Method** (6 levels including Vanilla). We report the effect size $\omega^2$, which measures the proportion of variance explained by each factor after accounting for others.

*Table 10.* Coalitions with the highest explanatory power for compatibility between accuracy and AOD on other datasets.

| Dataset | Mean Compatibility | Top-3 Supporting Coalitions | Top-3 Opposing Coalitions |
|---|---|---|---|
| UFRGS | $-0.0857 \pm 0.0225$ | SEX (27.34%)
LITERATURE (3.66%)
PORTUGUESE_ESSAY (1.92%) | PHYSICS, BIOLOGY, HISTORY, LITERATURE,
PORTUGUESE_ESSAY, CHEMISTRY (-0.16%)
HISTORY (-0.12%)
PHYSICS, BIOLOGY, HISTORY, LITERATURE,
PORTUGUESE_ESSAY, MATH, CHEMISTRY (-0.11%) |
| COMPAS | $0.0130 \pm 0.0605$ | $\emptyset$ (543.84%)
PRIORS_COUNT (21.00%)
AGE, DECILE_SCORE, C_CHARGE_DEGREE
(2.69%) | DECILE_SCORE (-158.23%)
AGE (-100.80%)
AGE, DECILE_SCORE, PRIORS_COUNT (-27.54%) |
| DIABETES | $-0.0552 \pm 0.0220$ | AGE (46.14%)
GLUCOSE (27.17%)
PREGNANCIES (11.97%) | BMI (-3.48%)
INSULIN (-1.01%)
SKINTHICKNESS (-0.69%) |
| BANK | $-0.0227 \pm 0.0240$ | POUTCOME (58.83%)
AGE (3.31%)
AGE, POUTCOME (2.90%) | PDAYS (-7.57%)
HOUSING (-6.96%)
MARITAL (-5.58%) |
| HEART | $0.1173 \pm 0.0951$ | $\emptyset$ (235.19%)
RESTING_ELECTROCARDIOGRAM, ST_SLOPE,
NUM_MAJOR_VESSELS (0.23%)
RESTING_ELECTROCARDIOGRAM, ST_SLOPE
(0.22%) | NUM_MAJOR_VESSELS (-23.10%)
THALASSEMIA (-22.23%)
SEX (-9.42%) |

Figure 7 visualizes the results, revealing three key patterns:

1. **Dominance of Dataset × Model Interactions for Fairness-Utility Pairs.** For Accuracy-AOD and Accuracy-CFVR, the interaction term Dataset × Model is the dominant factor ($\omega^2 = 0.557$ and $0.207$, respectively), surpassing any single factor. Compatibility is not an inherent property of a dataset or a model alone, but arises from their specific *interplay*. For example, a linear model (LR) may exhibit different compatibility patterns than a nonlinear model (MLP) on the same dataset due to differences in how they encode features.

2. **Method Dominance for Fairness-Fairness Pairs.** In contrast, for AOD-CFVR, the Method choice is the strongest predictor ($\omega^2 = 0.358$), suggesting that the alignment between different fairness definitions is largely determined by the debiasing strategy employed. Practitioners can influence Fairness-Fairness compatibility through method selection, but have less control over Fairness-Utility compatibility (which is more constrained by data and model architecture).

3. **Overall Dataset Impact.** Across all three pairs, the Dataset factor (either alone or in interactions) consistently plays a major role, highlighting the data-dependent nature of these trade-offs. This corroborates our finding in Appendix B.3 that compatibility is highly context-specific.

**Predictive Modeling Failure.** To test whether practitioners could predict compatibility without exhaustive evaluation, we trained regression models (Decision Tree, Random Forest, Gradient Boosting) using Leave-One-Dataset-Out (LODO) cross-validation. Features included dataset meta-statistics (Class Imbalance, Average Correlation, Mutual Information from Table 8), along with model and method types (one-hot encoded). All models failed to generalize, yielding negative $R^2$ scores (e.g., Decision Tree: $R^2 = -8.54 \pm 15.52$ for Accuracy-AOD). This failure is instructive: it demonstrates that compatibility is an *emergent property* arising from complex, nonlinear interactions between data, model, and method that cannot be captured by simple marginal statistics. There is no shortcut to estimating compatibility. Practitioners must perform empirical evaluation for each specific dataset-method-model combination. However, our framework makes this evaluation tractable by providing a unified metric (cosine similarity of representation vectors) and mechanistic explanations (coalition attribution).

### B.7. Coalitions with the Highest Explanatory Power on Other Datasets

In Table 6, we presented the top explanatory coalitions for the Census Income dataset. Here, we provide the corresponding results for the remaining five datasets: UFRGS, COMPAS, Diabetes, Bank Marketing, and Heart Disease. Tables 10, 11, and 12 detail the top-3 supporting and opposing coalitions for three representative metric pairs: Accuracy-AOD, Accuracy-CFVR, and AOD-CFVR.

The results reinforce the key findings observed in the main text:

*Table 11.* Coalitions with the highest explanatory power for compatibility between accuracy and CFVR on other datasets.

| DATASET | MEAN COMPATIBILITY | TOP-3 SUPPORTING COALITIONS | TOP-3 OPPOSING COALITIONS |
|---|---|---|---|
| UFRGS | -0.0794 ± 0.0205 | SEX (34.18%)
SEX, PHYSICS, HISTORY, GEOGRAPHY, LITERATURE, PORTUGUESE_ESSAY (0.57%)
SEX, PHYSICS, HISTORY, LITERATURE, PORTUGUESE_ESSAY (0.43%) | RACE, PORTUGUESE_ESSAY (-0.17%)
SEX, RACE, PHYSICS, BIOLOGY, HISTORY, SECOND_LANGUAGE, GEOGRAPHY, LITERATURE, PORTUGUESE_ESSAY, MATH, CHEMISTRY (-0.07%)
RACE, PHYSICS, BIOLOGY, HISTORY, SECOND_LANGUAGE, LITERATURE, PORTUGUESE_ESSAY, MATH, CHEMISTRY (-0.07%) |
| COMPAS | -0.0208 ± 0.0070 | AGE (85.75%)
SEX, PRIORS_COUNT (4.96%)
SEX, DECILE_SCORE (4.92%) | AGE, DECILE_SCORE (-16.53%)
AGE, DECILE_SCORE, C_CHARGE_DEGREE (-6.15%)
AGE, PRIORS_COUNT (-5.01%) |
| DIABETES | -0.0302 ± 0.0214 | AGE (97.92%)
GLUCOSE, AGE (5.32%)
INSULIN, AGE (3.10%) | GLUCOSE, INSULIN, AGE (-3.66%)
GLUCOSE, BMI, AGE (-2.23%)
BLOODPRESSURE, BMI, AGE (-1.17%) |
| BANK | -0.0146 ± 0.0125 | AGE, HOUSING (44.46%)
AGE (24.69%)
AGE, EDUCATION (8.88%) | AGE, JOB, EDUCATION (-3.42%)
AGE, HOUSING, PDAYS, POUTCOME (-1.91%)
AGE, JOB, MARITAL, EDUCATION, HOUSING (-1.56%) |
| HEART | -0.0225 ± 0.0234 | SEX (113.33%)
SEX, NUM_MAJOR_VESSELS (16.63%)
SEX, THALASSEMIA (3.80%) | AGE, SEX, CHEST_PAIN_TYPE, RESTING_BLOOD_PRESSURE, CHOLESTEROL, FASTING_BLOOD_SUGAR, RESTING_ELECTROCARDIOGRAM, MAX_HEART_RATE_ACHIEVED, EXERCISE_INDUCED_ANGINA, ST_DEPRESSION, ST_SLOPE, NUM_MAJOR_VESSELS, THALASSEMIA (-0.83%)
AGE, SEX, CHEST_PAIN_TYPE, RESTING_BLOOD_PRESSURE, FASTING_BLOOD_SUGAR, RESTING_ELECTROCARDIOGRAM, MAX_HEART_RATE_ACHIEVED, EXERCISE_INDUCED_ANGINA, ST_DEPRESSION, ST_SLOPE, NUM_MAJOR_VESSELS, THALASSEMIA (-0.71%)
AGE, SEX, CHEST_PAIN_TYPE, RESTING_BLOOD_PRESSURE, CHOLESTEROL, FASTING_BLOOD_SUGAR, RESTING_ELECTROCARDIOGRAM, EXERCISE_INDUCED_ANGINA, ST_DEPRESSION, ST_SLOPE, NUM_MAJOR_VESSELS, THALASSEMIA (-0.58%) |

- **Ubiquity of Sensitive Attributes in CFVR:** For pairs involving CFVR (Tables 11 and 12), the top coalitions almost exclusively contain sensitive attributes (e.g., sex, age, race), validating the causal nature of the metric.

- **Empty Set Impact:** The empty set $\emptyset$ frequently appears as a dominant factor, particularly for the Accuracy-AOD pair (Table 10), highlighting the pervasive role of label and group imbalance.

- **Low-Order Dominance:** Across all datasets, the most influential coalitions remain predominantly low-order, with high-order interactions typically contributing to noise.

## B.8. Robustness to Baseline Choices

The exact values of Harsanyi interactions depend on how masked features are instantiated. Our main experiments use the feature-wise training mean as a computationally efficient global baseline. However, for categorical variables encoded as integers, the mean can correspond to an out-of-distribution value. To assess whether our conclusions are artifacts of this choice, we compare the mean baseline against two semantically valid alternatives: the median baseline, which preserves integer-valued discretized features, and the mode baseline, which preserves valid categorical codes. The ablation is conducted on all six datasets with vanilla Logistic Regression and 10 random seeds.

Table 13 shows that the direction of metric interaction vectors is highly stable across baseline choices, with an average cross-baseline cosine similarity of 0.9607. The compatibility landscape is also preserved qualitatively: Utility–Utility pairs remain strongly aligned, Fairness–Fairness pairs remain synergistic, and Fairness–Utility pairs remain close to orthogonal. Therefore, our main geometric conclusions are not artifacts of the mean baseline. Nevertheless, baseline selection remains a limitation for exact interaction values and fine-grained coalition attribution.

## B.9. Distributional Masking with Marginal Sampling

To further address the out-of-distribution concern of fixed global baselines, we evaluate a distributional masking strategy based on marginal sampling. For each observed subset $S$, instead of replacing the masked coordinates $N \setminus S$ with a fixed

*Table 12.* Coalitions with the highest explanatory power for compatibility between AOD and CFVR on other datasets.

| DATASET | MEAN COMPATIBILITY | TOP-3 SUPPORTING COALITIONS | TOP-3 OPPOSING COALITIONS |
|---|---|---|---|
| UFRGS | $0.7821 \pm 0.0492$ | SEX (63.57%)
RACE (3.12%)
SEX, LITERATURE (0.58%) | RACE, SECOND_LANGUAGE, GEOGRAPHY, LITERATURE, PORTUGUESE_ESSAY, MATH, CHEMISTRY (-0.00%)
RACE, PHYSICS, HISTORY, SECOND_LANGUAGE, GEOGRAPHY, PORTUGUESE_ESSAY, MATH (-0.00%)
RACE, PHYSICS, SECOND_LANGUAGE, GEOGRAPHY, PORTUGUESE_ESSAY, MATH (-0.00%) |
| COMPAS | $0.5723 \pm 0.0784$ | AGE (70.83%)
SEX (6.05%)
SEX, AGE (4.74%) | SEX, RACE, DECILE_SCORE, PRIORS_COUNT (-0.42%)
SEX, RACE, DECILE_SCORE (-0.29%)
SEX, AGE, RACE, DECILE_SCORE, PRIORS_COUNT (-0.15%) |
| DIABETES | $0.7707 \pm 0.0527$ | AGE (71.47%)
BMI, AGE (5.29%)
GLUCOSE, AGE (4.40%) | GLUCOSE, INSULIN, AGE (-0.08%)
PREGNANCIES, GLUCOSE, SKINTHICKNESS, DIABETESPEDIGREEFUNCTION, AGE (-0.05%)
PREGNANCIES, GLUCOSE, BLOODPRESSURE, SKINTHICKNESS, BMI, DIABETESPEDIGREEFUNCTION, AGE (-0.04%) |
| BANK | $-0.0042 \pm 0.0752$ | AGE, HOUSING, PDAYS, POUTCOME (153.64%)
AGE, POUTCOME (148.95%)
AGE, PDAYS (148.24%) | AGE (-344.70%)
AGE, MARITAL (-136.66%)
AGE, HOUSING, LOAN (-41.51%) |
| HEART | $0.3209 \pm 0.0616$ | SEX (55.81%)
SEX, NUM_MAJOR_VESSELS (7.69%)
AGE (5.66%) | AGE, CHEST_PAIN_TYPE, ST_DEPRESSION, NUM_MAJOR_VESSELS (-0.15%)
AGE, SEX, RESTING_ELECTROCARDIOGRAM, EXERCISE_INDUCED_ANGINA, ST_SLOPE (-0.02%)
AGE, CHEST_PAIN_TYPE, NUM_MAJOR_VESSELS (-0.02%) |

*Table 13.* Vector-level consistency of metric interaction vectors under different baseline choices, measured by cosine similarity. Values are averaged over six datasets and 10 random seeds using vanilla Logistic Regression.

| METRIC | MEAN–MEDIAN | MEAN–MODE | MEDIAN–MODE | AVG. |
|---|---|---|---|---|
| ACCURACY | 0.9965 | 0.9956 | 0.9986 | 0.9969 |
| RECALL | 0.9814 | 0.9310 | 0.9567 | 0.9564 |
| FPR | 0.9434 | 0.9244 | 0.9757 | 0.9478 |
| SPD | 0.9761 | 0.9510 | 0.9712 | 0.9661 |
| EOD | 0.9602 | 0.9404 | 0.9630 | 0.9545 |
| PED | 0.9735 | 0.9408 | 0.9664 | 0.9603 |
| AOD | 0.9655 | 0.9381 | 0.9666 | 0.9567 |
| CFVR | 0.9601 | 0.9333 | 0.9616 | 0.9517 |
| GIFVR | 0.9646 | 0.9381 | 0.9652 | 0.9560 |
| AVG. | 0.9690 | 0.9436 | 0.9694 | 0.9607 |

baseline, we estimate

$$v_{\mathrm{marginal}}(S) = \mathbb{E}_{x'_{N \setminus S} \sim P(X_{N \setminus S})} \left[ f(x_S, x'_{N \setminus S}) \right] \approx \frac{1}{K} \sum_{k=1}^{K} f(x_S, x'^{(k)}_{N \setminus S}). \tag{19}$$

We use stratified marginal sampling with $K = 100$ strata per feature. This experiment is conducted on all six datasets using vanilla Logistic Regression and 10 random seeds.

Figure 8 shows that the central conclusions remain stable under marginal sampling. The compatibility landscape preserves the same qualitative hierarchy as the fixed-baseline setting: Utility–Utility pairs remain strongly aligned, Fairness–Fairness pairs remain synergistic, and Fairness–Utility pairs remain close to orthogonal. The interaction-order analysis also confirms that most high-order coalitions contribute negligibly, while low-order coalitions dominate the compatibility structure. Thus, our findings are not artifacts of out-of-distribution mean masking. The main limitation is computational: marginal sampling increases the cost of each masked evaluation by approximately a factor of $K$. Moreover, marginal sampling still assumes independent feature replacement; conditional sampling could better preserve feature dependencies but would introduce additional modeling and computational overhead.

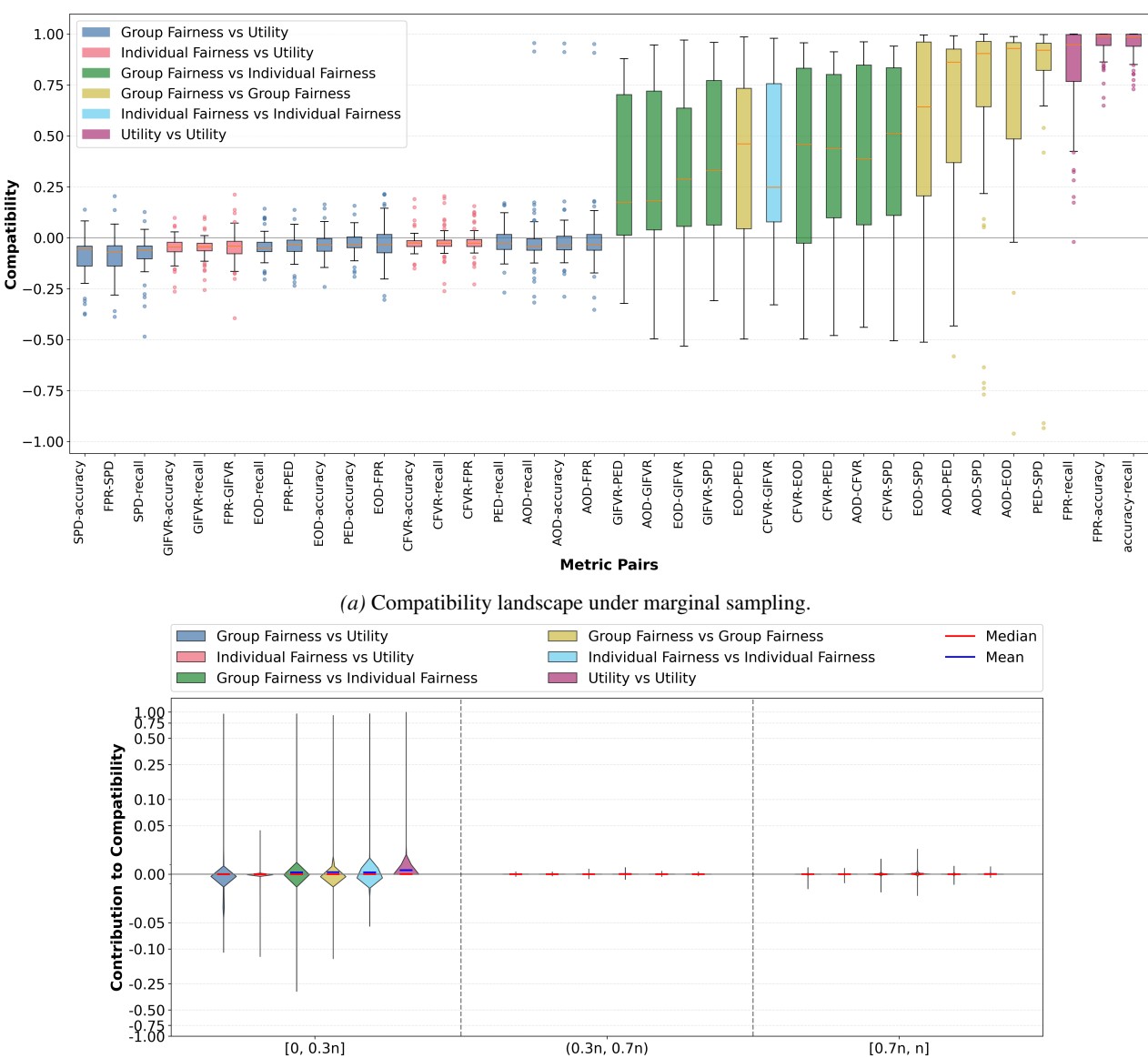

*(a)* Compatibility landscape under marginal sampling.

*(b)* Coalition contributions by interaction order under marginal sampling.

*Figure 8.* Robustness of compatibility analysis under distributional masking. Marginal sampling preserves the main hierarchy of metric relationships and confirms that compatibility is dominated by sparse, low-order coalitions.

# C. Discussion

In this section, we synthesize our findings to clarify the conceptual distinction between compatibility and trade-offs, and discuss practical implications for fairness practitioners. We also outline the limitations of our study and propose directions for future research. We emphasize that our framework is a diagnostic tool rather than a normative decision rule. It does not prescribe which fairness metric should be prioritized, nor does it eliminate trade-offs. Instead, it explains which coalition-level mechanisms make metric pairs aligned, orthogonal, or conflicting, thereby informing metric selection, intervention design, and Pareto analysis. Final metric choices remain application-dependent and should involve domain experts and affected stakeholders.

## C.1. Why Cosine Similarity Rather Than Simpler Alternatives?

Our choice of cosine similarity is driven by the diagnostic goal of compatibility analysis: determining whether two metrics are governed by similar coalition-level mechanisms. For this purpose, a compatibility measure should satisfy three desiderata. First, it should be bounded and sign-interpretable, so that positive values indicate aligned metric requirements, near-zero values indicate approximate orthogonality, and negative values indicate opposing requirements. Second, it should be invariant to positive rescaling, since different fairness and utility proxies can have different natural magnitudes. Third, it should support coalition-level attribution, allowing the overall compatibility score to be decomposed into the contributions of individual coalitions, as in Eq. 4. cosine similarity satisfies these requirements.

Several simpler alternatives are less suitable for this diagnostic goal. Pearson correlation also measures directional association, but it first centers each interaction vector by subtracting its mean. As a result, each coalition is evaluated relative to the average interaction over all coalitions, rather than through its original interaction value, making coalition-level attribution less direct. The unnormalized inner product preserves additive decomposition over coalitions, but it is sensitive to vector magnitude and unbounded, so metrics with larger proxy scales can appear more compatible even when their interaction directions are similar. Rank-based measures such as Spearman or Kendall correlation capture monotonic association, but they discard the signed magnitudes of coalition contributions and therefore cannot directly explain which coalitions produce synergy or conflict. Thus, cosine similarity is a conservative choice for our purpose: it compares structural direction while preserving an interpretable coalition-level decomposition.

This interaction-space perspective also differs from gradient- or first-order attribution-based diagnostics. Gradient alignment can indicate whether two optimization objectives locally agree in parameter space, but it does not identify which semantically meaningful feature coalitions explain the metric conflict. First-order attribution methods such as SHAP (Lundberg & Lee, 2017) or LIME (Ribeiro et al., 2016) assign importance to individual features, which is useful for marginal explanations but insufficient for intersectional fairness questions where conflicts may arise from joint patterns such as age–race or sex–occupation. Our interaction-space formulation is designed to preserve such higher-order coalitions, so it complements rather than replaces gradient- or attribution-based diagnostics.

## C.2. Compatibility vs. Trade-off: A Conceptual Clarification

A key conceptual contribution of this work is distinguishing between *compatibility* and the traditional notion of *trade-off*:

- **Trade-off (Outcome-based):** Conventionally, a trade-off is observed when improving metric $m_1$ necessarily degrades metric $m_2$ during optimization. This is a *post-hoc* observation of the optimization outcome, typically visualized via Pareto frontiers. Trade-offs are empirical facts about specific optimization trajectories.

- **Compatibility (Mechanism-based):** In contrast, we define compatibility as the cosine similarity between attribute-level interaction vectors of two metrics, computed by decomposing how a trained model's decision function $v(x)$ contributes to each metric. This characterizes the *structural alignment* of the model's learned representations for different metrics, revealing the underlying mechanism that drives or prevents trade-offs.

**Relationship.** Compatibility and Pareto trade-offs are related but not equivalent. Compatibility is measured within the interaction representation of a trained model, whereas a Pareto frontier is an outcome of a specific optimization procedure across multiple models or hyperparameters. Low compatibility can indicate that two metrics rely on conflicting or weakly shared mechanisms, making severe trade-offs more plausible. However, the realized optimization trajectory also depends on loss landscape curvature, model capacity, constraints, and optimizer dynamics. Thus, compatibility should be used as a structural diagnostic rather than a direct and deterministic predictor of convergence or Pareto shape.

**Complementarity with Pareto Analysis.** Compatibility does *not* replace Pareto analysis:

- **Pareto analysis:** *What* trade-offs exist (outcome-oriented, requires multiple models).

- **Compatibility analysis:** *Why* trade-offs exist (mechanism-oriented, requires one model).

Together, compatibility suggests where joint optimization may be difficult, while Pareto analysis measures the realized empirical cost.

### C.3. Practical Implications

Our findings offer actionable guidelines for fairness-aware ML development:

**1. Metric Selection: Prioritize Compatible Pairs.** Use compatibility scores (Figure 1) to identify naturally aligned metric combinations. For example, FPR and AOD exhibit positive median compatibility, making them a coherent pair for group fairness evaluation—optimizing one tends to improve the other. In contrast, Accuracy and SPD show negative compatibility, requiring careful trade-off management (e.g., Pareto-based multi-objective optimization). *Caveat:* Compatibility is context-dependent (Appendix B.3, B.6). Practitioners should compute compatibility on their specific data before finalizing metric selection.

**2. Method Selection: Match Interventions to Conflict Sources.** Use coalition attribution (Tables 6, 10, 11, and 12) to diagnose the source of compatibility issues:

- **If ∅ (empty set) dominates:** Dataset-level biases (label/group imbalance) drive conflicts ⇒ Use pre-processing (e.g., Reweighing, Fair-SMOTE).

- **If low-order coalitions with sensitive attributes dominate** (e.g., {sex}, {age, race}): Simple attribute interaction-level effects drive conflicts ⇒ Use in-processing methods that model these interactions (e.g., HIFI, adversarial debiasing).

- **If high-order coalitions dominate:** Complex entangled patterns drive conflicts ⇒ Consider feature engineering or ensemble methods (e.g., MAAT).

Additionally, choose methods based on tolerance for utility loss: Fair-SMOTE achieves large fairness gains but compresses compatibility space significantly (Table 7); MAAT preserves utility compatibility while achieving modest fairness gains.

**3. Hyperparameter Tuning: Monitor Compatibility as a Diagnostic Heuristic.** Track compatibility during training to identify candidate debiasing regimes before substantial performance degradation occurs. Proposed protocol:

1. Compute baseline compatibility $\mathcal{C}(m_1, m_2)$ for critical metric pairs on a validation set.

2. As debiasing strength increases (e.g., regularization $\eta$), recompute compatibility at each checkpoint.

3. Identify the "sweet spot" where fairness-related compatibility drops (effective debiasing) but Utility-Utility compatibility remains high (preserved task structure).

4. In this case study, stopping before Utility-Utility compatibility collapses helps avoid the regime where decision-boundary distortion coincides with performance degradation.

In our HIFI case study (Figure 3), the sweet spot is $\eta < 1000$ (AOD and CFVR improve, Utility-Utility $> 0.9$). At $\eta \geq 1000$, Utility-Utility collapses to $\sim 0.6$, coinciding with the Pareto elbow. *Caveat:* This protocol is validated on one case study; practitioners should verify its effectiveness on their specific setup.

### C.4. Limitations and Future Work

C.4.1. LIMITATIONS

We acknowledge several limitations of our current framework:

1. **Computational Complexity.** Exact Harsanyi decomposition requires $O(2^{|N|})$ masked evaluations per sample. Without approximation, this restricts direct use to low-dimensional tabular datasets; in our implementation, a realistic upper bound is approximately $|N| \leq 15$ features, depending on the number of samples and metrics. *Mitigation strategies:*

   - **Feature engineering or dimensionality reduction:** Reduce $|N|$ using domain knowledge, feature selection, or PCA before compatibility analysis (Meng et al., 2025; Ren et al., 2025).
   - **Sparse interaction approximation:** Estimate interactions from sampled coalitions by exploiting the empirical sparsity of Harsanyi interactions (Ren et al., 2023a; 2024; Li & Zhang, 2023).
   - **Neural or amortized approximation:** Use learned approximators to estimate interaction effects more efficiently in high-dimensional settings (Chen et al., 2023a).

2. **Tabular Data Only.** Our empirical study focuses on tabular datasets with binary classification. While this covers many high-stakes domains (finance, healthcare, criminal justice), the patterns may differ for:

   - **Unstructured data:** Images and text have different interaction structures (e.g., spatial/sequential dependencies).
   - **Multi-class/regression tasks:** Fairness metrics for these tasks require different continuous proxies.

3. **Continuous Proxies.** We rely on continuous approximations of discrete metrics (Table 3), which are highly correlated with their discrete counterparts (Table 4) but can introduce proxy-specific distortions.

   - Our proxies cannot directly decompose metrics involving *ratios or non-linear functions of expectations* (e.g., Precision, F1-score, MCC, ROC-AUC), limiting the scope of the current formalization.
   - **Threshold effect:** If an intervention moves many predictions slightly across the decision boundary, discrete metrics may change sharply due to label flips, while continuous proxies register only small probability changes.
   - **Calibration effect:** If an intervention mostly changes confidence without changing labels, continuous proxies may change substantially while discrete metrics remain unchanged.
   - Therefore, proxy-induced error becomes practically meaningful when model updates mainly affect confidence rather than class assignments, or class assignments rather than confidence.

4. **Generalizability of Findings.** Our empirical findings (e.g., "orthogonality dominates", "low-order interactions drive compatibility") are based on six benchmark datasets. While these datasets are widely used in fairness research, they may not represent all application domains. Practitioners should validate our findings on their specific data before drawing conclusions.

### C.4.2. FUTURE WORK

We identify several promising directions for extending this work:

1. **Efficient Approximation Algorithms.** Develop sampling-based, sparsity-based or neural approximation methods for Harsanyi interactions in high-dimensional settings, building on recent work in interaction-based XAI (Ren et al., 2023a; Li & Zhang, 2023; Ren et al., 2023b; Chen et al., 2023a).

2. **Extension to Unstructured Data.** Apply the compatibility framework to image and text domains, where fairness concerns are increasingly prominent. This requires addressing three barriers beyond the tabular setting studied here.

   - **Semantic barrier:** Raw pixels or tokens do not always correspond to independent fairness-relevant attributes. Applying our framework requires a principled mapping from raw inputs to meaningful concepts, such as image regions, superpixels, text spans, or learned concept bases.
   - **Computational barrier:** These domains involve many more input units than our current exact decomposition can handle, requiring sparse, sampling-based, or neural approximations to Harsanyi interactions.
   - **Metric barrier:** Domain-specific fairness and utility metrics may require new continuous proxies before interaction-level decomposition is possible.

3. **Theoretical Foundations.** Formalize the mathematical relationship between compatibility and Pareto frontiers. Specifically:

   - *Conjecture:* The curvature of the Pareto frontier is inversely related to compatibility (low total compatibility $\Rightarrow$ sharp curvature $\Rightarrow$ steep trade-offs).

     • Prove conditions under which compatibility serves as a lower/upper bound on achievable trade-offs.

4. **Non-linear Metric Decomposition.** Extend compatibility analysis to metrics defined as ratios or non-linear functions of multiple expectations, such as F1-score, MCC, and ROC-AUC. A possible route is to decompose the underlying linear components first and then approximate the non-linear metric via local linearization or other sensitivity-based approximations. This extension would introduce additional approximation error and computational overhead, so we leave a formal treatment to future work.

5. **Compatibility-Aware Optimization.** Develop optimization algorithms that explicitly maximize the alignment of representation vectors between fairness and utility objectives. Potential approaches: Adjust loss weights based on real-time compatibility monitoring during training.

6. **Interactive Tools for Practitioners.** Build software tools that:

     • Automatically compute compatibility for user-specified metric pairs.
     • Visualize coalition attribution to guide intervention design.
     • Monitor compatibility during training and alert users to potential trade-offs.

