# OpenReview forum: "A Game-Theoretic Framework for Measuring and Explaining Metric Compatibility in Fair Machine Learning"
_ICML.cc/2026/Conference — ICML 2026 regular_

### Official Review · Reviewer_CfpE · 2026-03-04

**Soundness:** 3
**Presentation:** 2
**Significance:** 2
**Originality:** 3
**Overall Recommendation:** 4
**Confidence:** 4

**Summary:**

The paper describes a framework for measuring and explaining the compatibility between fairness and utility metrics in binary classification. The central idea is not to look only at the final result (e.g., a Pareto frontier between accuracy and fairness), but to study whether two metrics are structurally aligned or in conflict within the model itself. To do this, they decompose each metric into a vector of contributions from attribute coalitions using Harsanyi interactions, and then define the compatibility between two metrics as the cosine similarity between those vectors.
Since many standard metrics are discrete and cannot be directly decomposed, the authors first introduce continuous approximations based on predicted probabilities, only including metrics that can be expressed  as linear expectations (e.g., SPD, EOD, PED, AOD, GIFVR, CFVR, accuracy, recall, FPR), and leave out metrics such as precision, F1, MCC, or ROC-AUC because they involve nonlinear operations. These approximations are validated by showing high correlations with the original discrete metrics.
The approach introduced in the paper was applied on 6 datasets, 7 models, 6 debiasing methods, and 9 metrics, with 10 seeds and 500 test samples to calculate interactions for efficiency. Their main finding is that the fairness–utility relationship is usually one of near orthogonality (compatibility close to 0) rather than outright opposition (close to −1). This suggests that, in many cases, improving fairness is not  directly in opposition  with utility, but rather that both depend in part on different mechanisms.
The authors  also find that compatibility is dominated by rare, low-order interactions (individual or pair attributes), and that the empty set captures baseline biases in the dataset (class or group imbalance). Furthermore, they show that debiasing methods do not eliminate the conflict, but rather compress the compatibility space.

**Compliance With Llm Reviewing Policy:**

Affirmed.

**Final Justification:**

The authors propose an interesting idea switching the usual focus of classical methods addressing utility-fairness trade-offs. As such, I recognize the novelty and originality of the approach. Moreover, I acknowledge the authors' efforts in the rebuttal, and I am thankful for their response which leaves no lingering questions. However, some concerns remain on the ultimate scope of the proposed method (it does not address common metrics like F1 and AUC) and on its scalability.

In light of this, I believe my initial assesment was fair, and my recommendation is a **weak accept**.

**Key Questions For Authors:**

It would be important to answer these questions
1. Has a more direct comparison been made with simpler alternatives (e.g., correlations between gradients or local sensitivity)?
2. Has an assessment been made of the robustness of compatibility with different baseline and discretization choices?
3. Has at least one preliminary experiment with higher dimensionality been conducted?.

**Limitations:**

Yes

**Strengths And Weaknesses:**

Strengths:
The real contribution, in my opinion, is that the paper changes the question: it doesn't just ask “is there a trade-off between fairness and accuracy?”, but “why” two metrics clash or align. This is different from the classic approach of theoretical impossibility or empirical Pareto.
It introduces an operational definition of compatibility between metrics by representing each metric as a vector of interactions and comparing two metrics with cosine similarity. That is the central methodological innovation.
The paper also  provides mechanistic attribution,  not just a compatibility scalar, but which coalitions of attributes explain the synergy or conflict. This makes it possible to distinguish whether the problem stems from sensitive attributes, simple combinations, or the overall imbalance of the dataset.
Finally, the authors  extend HIFI, which  already used Harsanyi interactions to decompose bias in group fairness metrics. The work extends it to several fairness and utility metrics and uses the decomposition to study relationships between metrics, not just an isolated metric.
Under my perspective, the most solid contribution is not discovering that fairness and utility are orthogonal , but rather having proposed an interpretable geometry of metrics that connects decomposition, similarity, and attribution.

Weaknesses
The main limitation is that the method relies on continuous approximations and only covers metrics that are linear expectations, leaving out important metrics such as precision, F1, MCC, or ROC-AUC. In addition, the exact decomposition has a cost of O(2^{|N|}) per sample, which greatly restricts scalability. Empirical evidence focuses on tabular data and binary classification, and it is unclear how much of this holds true in more realistic or high-dimensional scenarios more common in real problems.  Schematically, the three weaknesses found are:   (1) It does not solve the problem of which fairness metric to choose. (2) It does not eliminate trade-offs since it only offers a new way to measure and explain them. (3) The empirical part is useful, but the general conclusions are limited to small/medium tabular datasets and linearizable metrics.

The novelty is moderate to high. The paper does not invent Harsanyi interactions in fairness; rather, it builds on HIFI and extends it in three ways: more metrics, inter-metric analysis, and connection with debiasing/Pareto dynamics. The new twist that the authors give to Harsanyi's decomposition seems substantial to me, although incremental with respect to the technical substrate.

The formulation is well motivated, and the construction that evolves from continuous proxy to interaction vector and cosine distance is consistent. The validation of proxies with significant correlations helps. Even so, there is a conceptual leap between geometric compatibility and actual optimization difficulty. The paper acknowledges this, but at times the text suggests a stronger causal or dynamic interpretation than the experiments demonstrate.

The experimental battery is extensive for a methodological paper. The results on metric clustering, dataset dominance, and predictive model failure reinforce the idea that compatibility is context-dependent.

---

> ### Author Rebuttal · Authors · 2026-03-29
>
> We thank the reviewer for the thorough assessment. We clarify our scope, limitations, and design choices below:
>
> ---
>
> **Addressing Scope Limitations (Weaknesses 1, 2, 3)**
>
> We agree our framework neither prescribes metric choices nor eliminates trade-offs. It is a *mechanistic diagnostic tool* explaining *why* metrics conflict at the representation level to inform practitioners. We also acknowledge the limitation to linearizable metrics.
>
> ---
>
> **Q1: Geometric Compatibility vs. Simpler Alternatives**
>
> We chose cosine similarity of Harsanyi interactions because fairness-utility trade-offs require **feature-level and intersectional** explanations.
>
> 1. **Vs. Gradient Alignment (Parameter Space):** Gradients show *if* metrics conflict but cannot explain *why* semantically (e.g., "Is it driven by Age?"). Our framework operates in the *interaction space*, attributing compatibility directly to feature coalitions.
> 2. **Vs. First-order Attribution (SHAP/LIME):** Fairness is inherently intersectional (e.g., "Young Black Females"). First-order methods capture marginal importance but fail to isolate joint coalition effects. Harsanyi interactions explicitly disentangle these higher-order entanglements.
>
> We will add this comparative justification to Appendix C.
>
> ---
>
> **Q2: Baseline Robustness and Discretization Choices**
>
> Thank you for raising this. We evaluated baseline robustness across **all 6 datasets** (Vanilla LR, 10 seeds), comparing **Mean**, **Median**, and **Mode** baselines (the latter two ensuring in-distribution values for discretized features).
>
> Results (`https://icml2026authorresponse-paper808.netlify.app/`) show robustness at two levels:
>
> *Representation Vector-level Consistency:* Cosine similarity between metric decomposition vectors from different baselines averages **0.96** (Mean↔Median **0.969**, Mean↔Mode **0.944**, Median↔Mode **0.969**). The **geometric structure in coalition space is stable**.
>
> *Landscape-level Robustness:* The hierarchical structure (Utility-Utility alignment, Fairness-Fairness synergy, Fairness-Utility near-orthogonality) remains consistent with Figure 1. Macroscopic metric relationships are **not artifacts of the mean baseline**.
>
> **Discretization Robustness:** While lacking a systematic ablation on granularity, our preprocessing preserves data distributions (using domain thresholds or balanced binning). The consistency of compatibility patterns across six diverse datasets—each with different discretization characteristics—suggests robustness to reasonable discretization choices.
>
> We will add this analysis to Appendix C.
>
> ---
>
> **Q3: Higher Dimensionality and Scalability**
>
> We agree that the $\mathcal{O}(2^{|N|})$ complexity currently limits exact computation to datasets with $|N| \le 15$.
>
> Due to this bottleneck, we focus on tabular data. This practically encompasses many high-stakes fairness applications (e.g., lending, criminal justice) where features possess explicit semantic meanings (e.g., *Age*, *Race*) essential for coalition-level attribution.
>
> Extending to high-dimensional unstructured data (images/text) involves two challenges beyond this foundational work's scope:
>
> 1. **The Semantic Barrier:** Raw pixels/tokens lack independent fairness semantics. Applying our framework requires a robust mapping to high-level concepts (e.g., superpixels, PCA features) as base attributes.
> 2. **The Computational Barrier:** Scaling requires advanced sparsity-based Harsanyi approximations (Ren et al., 2023, CVPR; Ren et al., 2023, ICLR).
>
> To prevent overclaiming, we will explicitly define this scope ($|N| \le 15$) and formalize the roadmap for unstructured data in the Limitations and Future Work sections.

---

> > ### Author Rebuttal · Reviewer_CfpE · 2026-04-04
> >
> > We thank the authors for their rebuttal. Our concerns have been adressed and we have no further questions, so we will keep our positive score.

---

### Official Review · Reviewer_RWsk · 2026-03-06

**Soundness:** 3
**Presentation:** 4
**Significance:** 3
**Originality:** 3
**Overall Recommendation:** 4
**Confidence:** 3

**Summary:**

This paper introduces a game-theoretic framework to measure and explain the compatibility between fairness and utility metrics in machine learning. The authors decompose metrics into interaction vectors using Harsanyi interactions and measure their compatibility using cosine similarity. The empirical results show that fairness and utility are often structurally orthogonal.

**Compliance With Llm Reviewing Policy:**

Affirmed.

**Key Questions For Authors:**

1. How does the proposed method scale to non-linear industrial metrics, and what mathematical extensions are required?
2. Have the authors considered utilizing approximation methods to deal with the exponential time complexity associated with Harsanyi interaction calculations?

**Limitations:**

yes

**Strengths And Weaknesses:**

**Strength**: The paper is well-written and easy to follow. The proposed framework requires no causal assumption nor prior knowledge of causal relationships, making it easier to apply in real-world applications. The authors conduct comprehensive experiments on 6 datasets, 7 models, and 6 debiasing methods.

**Weaknesses**:
1. The framework relies on continuous probability scores to approximate discrete labels. However, it is unable to handle non-linear metrics, which are frequently critical in industrial applications.
2. It is known that calculating Harsanyi interaction takes exponential time. This significantly limits the framework's efficiency when applied to large-scale machine-learning tasks.

---

> ### Author Rebuttal · Authors · 2026-03-29
>
> We sincerely thank the reviewer for the positive feedback and for highlighting the practical value of our oracle-free framework. We address your valid concerns regarding metric scope and scalability:
>
> ---
>
> **W1 & Q1: Inability to Handle Non-linear Metrics**
>
> We explicitly acknowledge this as a **theoretical boundary** of our current framework (as noted in Appendix C.3.1), not merely a computational one.
>
> 1. **The Theoretical Barrier**
>
>   Our framework relies on Harsanyi dividends, which require metrics to be expressible as **linear expectations** of model outputs. Non-linear metrics like F1 are typically ratios or non-linear functions of multiple expectations. This violates the additivity required for direct coalition-level decomposition, which is why Table 3 explicitly excludes them.
>
> 2. **Practical Workaround: Proxy-Based Diagnosis**
>
>   Rather than direct decomposition, we suggest a practical **two-stage evaluation strategy** for real-world applications:
>
>   - **Stage 1 (Mechanistic Diagnosis):** Use our framework on linear proxy metrics (e.g., Accuracy, Recall, FPR). As shown in Table 4, our continuous proxies achieve excellent correlation with their discrete counterparts. This makes them highly reliable surrogates for identifying which attribute coalitions drive metric conflicts.
>   - **Stage 2 (Final Validation):** Evaluate the final model using the strict non-linear metrics (F1, MCC) on a hold-out set to ensure industrial KPIs are met.
> 3. **Potential Mathematical Extensions**
>
>   Extending our framework to non-linear metrics requires bridging the gap between non-additive functions and our additive Harsanyi decomposition. Since non-linear metrics are constructed from linear base components, a potential mathematical extension would entail first applying Harsanyi decomposition to these individual linear components, and subsequently using approximation techniques (e.g., local linearization) to estimate the overall coalition contributions. However, because this inherently introduces approximation errors and multiplies the computational overhead, we currently rely on the proxy-based approach for linear metrics and have carefully scoped the formalization of non-linear metric decomposition as future work.
>
>
> ---
>
> **W2 & Q2: Exponential Time Complexity and Approximations**
>
> We fully acknowledge that the $\mathcal{O}(2^{|N|})$ complexity limits exact computation to lower-dimensional tabular data ($|N| \le 15$). However, our compatibility framework is structurally agnostic to how the interactions are computed. For industrial applications with higher-dimensional features, it can directly integrate with recent Harsanyi approximation algorithms, including:
>
> - Sparsity-regularized learning (Ren et al., 2023, CVPR; Ren et al., 2023, ICLR)
> - PCA-based dimensionality reduction (Ren et al., 2025, AAAI)
>
> We will further expand our Limitations section to explicitly discuss this computational bottleneck and detail how these approximation methods can be leveraged to scale the framework to large-scale machine learning tasks in future work.

---

> > ### Author Rebuttal · Reviewer_RWsk · 2026-04-04
> >
> > Thanks for your reply, which fully solve my questions.

---

### Official Review · Reviewer_tauZ · 2026-03-13

**Soundness:** 3
**Presentation:** 3
**Significance:** 3
**Originality:** 3
**Overall Recommendation:** 4
**Confidence:** 3

**Summary:**

This paper studies the relationship between fairness and utility metrics through the lens of metric compatibility. It decomposes each metric into Harsanyi interaction vectors and defines compatibility as the cosine similarity between them, using this framework to analyze when fairness and utility are structurally aligned or misaligned. The main empirical message is that fairness-utility pairs often appear closer to near-orthogonality than direct opposition, and that this relationship is often driven by sparse, low-order coalitions. The paper also examines how debiasing methods reshape these compatibility structures. Overall, the paper can be viewed as extending HIFI-style decomposition into a broader framework for interpreting relationships among metrics.

**Compliance With Llm Reviewing Policy:**

Affirmed.

**Final Justification:**

I continue to view this as a solid paper with a clear and useful analytical perspective on fairness–utility relationships. The rebuttal clarified the intended scope of the claims and appropriately toned down some stronger interpretations, especially in Section 4.4. However, it did not add enough new evidence to change my overall assessment, so I keep my score unchanged.

**Key Questions For Authors:**

1. The paper defines compatibility as cosine similarity between metric interaction vectors, which is intuitive and interpretable. However, it remains unclear how strongly this structural notion should be expected to relate to optimization-level independence or weak practical trade-offs. Could the authors clarify the intended scope of this interpretation, and, if possible, provide additional evidence or discussion connecting compatibility patterns to actual training behavior?

2. In Section 4.4, the case study is interesting, and the coincidence with the Pareto elbow is suggestive. Still, it is not yet clear to me whether reduced utility–utility compatibility should generally be read as impending utility degradation. Could the authors clarify the mechanism behind this interpretation, and discuss whether they have observed similar behavior in other methods, models, or datasets?

3. How robust are the main conclusions to the use of continuous proxy metrics, and what kinds of distortion can these proxies introduce? The paper acknowledges that the framework relies on continuous approximations of discrete metrics and excludes several commonly used metrics such as Precision, F1-score, MCC, and ROC-AUC from the decomposition pipeline. Could the authors discuss more concretely when proxy-induced errors might become practically meaningful, and whether the main conclusions (e.g., near-orthogonality, low-order dominance) remain stable under alternative proxy choices?

**Limitations:**

yes

**Strengths And Weaknesses:**

Strengths

1. The paper studies, in a systematic way, the structural alignment and conflict among commonly used fairness and utility metrics in ML. In particular, the finding that fairness-utility relations are often closer to near-orthogonality than direct opposition is interesting. It could offer a useful perspective for model optimization or evaluation.

2. The paper is well organized, and one of its strengths is that the empirical observations and their broader interpretations are presented in a fairly structured way. The compatibility framework, coalition-level decomposition, and subsequent empirical claims are introduced in a progression that is easy to follow.

3. By showing that debiasing methods can reshape the compatibility structure among metrics, the paper provides a useful perspective for understanding what different debiasing methods are actually doing and what kinds of caution may be needed when selecting them.

Weaknesses

1. The main conclusions are established with respect to the paper’s specific definition of compatibility, namely cosine similarity between interaction vectors. While this is a reasonable and interpretable formulation, it does not by itself show that near-orthogonality translates directly into optimization-level independence or consistently weak practical trade-offs. This gap between structural compatibility and actual training behavior could be discussed more explicitly.

2. In Section 4.4, the case study is interesting, but the connection between collapsing utility-utility compatibility and actual performance degradation is not yet fully established. Also, the early-warning interpretation is supported only in a limited setting, and it remains unclear how broadly this phenomenon should be expected to hold.

3. The framework depends on continuous proxy metrics that are chosen for decomposability, which narrows the scope of the analysis. In particular, the paper excludes several commonly used utility metrics such as precision, F1-score, MCC, and ROC-AUC because they are not linear expectations and therefore do not fit the proposed decomposition pipeline. Although the proxy validation is helpful, this still limits how broadly the notion of “metric compatibility” can be interpreted in practice.

---

> ### Author Rebuttal · Authors · 2026-03-29
>
> We thank the reviewer for the thoughtful assessment, the positive evaluation of our structured presentation, and for recognizing the value of our empirical findings. We address your core questions below:
>
> ---
>
> **W1 & Q1: Structural Compatibility vs. Actual Optimization Dynamics**
>
> You raise a critical distinction. We want to clarify that our framework provides a *structural diagnostic* of metric alignment, not a deterministic predictor of optimization trajectories.
>
> Near-orthogonality ($C \approx 0$) suggests that the metrics rely on quasi-independent feature coalition subspaces. This structurally *allows* for joint optimization without severe direct gradient cancellation, explaining why constrained optimization *can* succeed. However, actual training behavior is fundamentally influenced by additional factors: loss landscape curvature, model capacity, and optimizer dynamics.
> In the revision, we will explicitly state this boundary in Section 4.2: compatibility explains *why* trade-offs are structurally mild or severe at the representation level, but it does not guarantee specific Pareto frontier shapes or optimization convergence.
>
> ---
>
> **W2 & Q2: Early Warning Case Study (Section 4.4)**
>
> We completely agree with your assessment. The observation that collapsing utility-utility compatibility precedes performance degradation is currently a preliminary finding demonstrated in a limited setting (HIFI on LR, Census Income).
>
> To carefully avoid overstating our claims, we will frame this as a "context-specific empirical observation" rather than a generalized "leading indicator". Specifically, we will revise Section 4.4 to explicitly state: This sequential collapse is currently observed primarily in the HIFI+LR+Census Income setting. While structurally suggestive, we do not claim this as a universal early-warning signal for performance degradation without broader validation across diverse tasks and architectures.
>
> ---
>
> **W3 & Q3: Dependence on Continuous Proxy Metrics**
>
> 1. **Robustness of Main Conclusions**
>
> Our core findings reflect the geometric and topological properties of the model's representation space, rather than artifacts of specific metric formulations. Therefore, they remain stable under reasonable proxy choices:
>
>
> - **Near-Orthogonality:** Metric compatibility is defined via cosine similarity, which captures the angular alignment of interaction vectors and is invariant to positive scaling. The near-zero median compatibility indicates that fairness and utility occupy largely orthogonal coalition-level representation subspaces. As long as the proxy preserves the ordinal ranking of model performance (supported by strong correlations, in Table 4), this geometric orthogonality remains structurally stable.
>
> - **Low-Order Dominance:** The concentration of interaction contributions in low-order coalitions (Figure 2) is a fundamental property of how models encode features (Ren et al., 2024, ICLR), not a byproduct of the metric definition. Transitioning from discrete to continuous formulations does not alter the underlying sparsity of the model's learned concepts.
> 2. **When Do Proxy-Induced Errors Become Meaningful?**
>
> We acknowledge that distortions arise precisely because continuous proxies measure *probabilistic confidence* while discrete metrics measure *hard threshold crossings*. This fundamental mathematical difference (smooth expectation vs. step function) manifests in two interconnected ways:
>
>
> - **Insensitivity to Margin Shifts (The Threshold Effect):** If a debiasing intervention shifts a sample's predicted probabilities slightly across the decision boundary (e.g., from 0.49 to 0.51), the discrete metric registers a massive change (a full label flip), whereas the continuous proxy registers only a negligible 0.02 shift.
>
> - **Oversensitivity to Confidence Scaling (The Calibration Effect):** Conversely, if an intervention shifts probabilities from 0.60 to 0.99, the discrete metric remains entirely unchanged (the label is still positive). However, the continuous proxy will heavily weight this increase in confidence. Consequently, if a model is severely miscalibrated, the proxy might indicate a distortion in utility or fairness that the discrete metric completely ignores.
>
>
> In summary, proxy distortion occurs when model updates primarily affect the *confidence* of predictions rather than their *class assignments* (or vice versa).
>
>
> We will add the above statements to Appendix C. Discussion.

---

> > ### Author Rebuttal · Reviewer_tauZ · 2026-04-04
> >
> > Thanks for the thoughtful responses. The rebuttal addressed my main concerns and appropriately narrowed several interpretations, especially regarding optimization behavior and the Section 4.4 case study. However, it primarily clarifies the intended scope of the claims rather than providing new supporting evidence. As a result, I keep my overall evaluation unchanged.

---

### Official Review · Reviewer_sZbz · 2026-03-13

**Soundness:** 2
**Presentation:** 3
**Significance:** 2
**Originality:** 3
**Overall Recommendation:** 5
**Confidence:** 3

**Summary:**

Machine learning fairness relies on numerous metrics — group fairness, individual fairness, and utility measures — that can conflict, agree, or behave independently. This paper proposes a game-theoretic framework to quantify metric compatibility by decomposing each metric into a vector of attribute-interaction contributions via Harsanyi dividends and measuring pairwise alignment through cosine similarity. Unlike causal approaches, the framework requires no causal graph, working directly from model predictions and data. An empirical evaluation across 6 datasets, 7 models, and 6 debiasing methods finds that fairness and utility metrics are predominantly orthogonal rather than sharply opposed, with compatibility driven by sparse low-order interactions.

**Compliance With Llm Reviewing Policy:**

Affirmed.

**Final Justification:**

Authors constructively answered most of the questions in the first phase of discussion, then after the second part conducted some additional evaluations addressing my concerns. I light of the strong rebuttal I decided to increase the score to recommend accept.

**Key Questions For Authors:**

Q1. Could the framework be generalized to use distributional masking (e.g., marginal or conditional sampling) instead of fixed baseline replacement?

Q2. How does the method scale with the number of features? What is a realistic maximum number of features that can be handled?

**Limitations:**

yes

**Strengths And Weaknesses:**

# Strengths:

**S1.** The paper is well-motivated and clearly written. The problem of understanding metric relationships in fairness-aware ML is presented convincingly, and the paper is overall easy to follow despite the technical density of the game-theoretic machinery.

**S2.** The proposed framework is quite general — it operates directly from model predictions and data without requiring causal graphs or assumptions about unbiased labels, which lowers the barrier to adoption compared to causal fairness approaches.

**S3.** The continuous metric approximations (Table 3) are a useful methodological contribution, enabling fine-grained interaction analysis where discrete metrics would not permit it. The validation of these proxies against original metrics (Table 4) strengthens confidence in this design choice.

**S4.** The empirical evaluation is thorough, spanning 6 datasets, 7 model architectures, and 6 debiasing methods with 10 random seeds each, yielding over 2,700 experimental configurations. This breadth lends credibility to the generality of the findings.

**S5.** The application of the framework to explain how debiasing methods reshape metric relationships (Section 4.3) is a compelling use case. The finding that debiasing compresses the compatibility space — reducing both synergies and conflicts — rather than simply eliminating trade-offs provides mechanistic insight into the inner workings of these methods that goes beyond standard benchmark comparisons.

# Weaknesses:

**W1.** The authors state they care about vector direction rather than magnitude, but do not discuss what formal properties a compatibility measure should satisfy in fairness-critical settings. No comparison with alternatives (Pearson correlation, inner product, rank-based measures) is provided.
**W2.** The framework replaces masked features with their training-set mean. This is not just an implementation detail — Proposition 3.2 structurally depends on the global baseline assumption. However, Ren et al. (2021), which the authors cite, showed that standard baseline masking (including mean baselines) often fails to faithfully represent feature absence. No robustness analysis with alternative masking strategies is provided, leaving it unclear how sensitive the compatibility scores and coalition attributions are to this foundational choice.

**W3.** The preprocessing converts categorical features to numerical codes, discretizes continuous features into bins, then uses the feature-wise mean as baseline. For categorical attributes like occupation or race, the mean of arbitrary codes (e.g., 1.4) has no semantic meaning and produces out-of-distribution inputs.

---

> ### Author Rebuttal · Authors · 2026-03-29
>
> We sincerely thank the reviewer for recognizing our framework's generality. We address your insightful concerns below:
>
> ---
>
> **W1: Choice of Compatibility Measure and Alternatives**
>
> We selected cosine similarity for three reasons (to be formalized in Sec 3.3):
> (1) **Bounded Interpretability:** It bounds compatibility in [-1, 1] (synergy, orthogonality, conflict).
> (2) **Scale-Invariance & Directionality:** It focuses on the structural alignment of coalition dependencies, not absolute magnitudes.
> (3) **Decomposability:** It enables straightforward coalition-level attribution (Eq. 4).
>
> Alternatives fall short:
> - **Pearson correlation** is magnitude-sensitive and non-decomposable.
> - **Inner product** is unbounded and lacks normalized geometric interpretation.
> - **Rank-based measures (Spearman/Kendall)** ignore underlying interaction structures and directional opposition.
>
> We will add this comparative discussion to the Appendix C.
>
> ---
>
> **W2, W3: Baseline Masking Strategy and OOD Issues**
>
> We agree that applying a mean baseline to categorical features introduces OOD samples. We initially chose the training-set mean as an engineering heuristic for computational efficiency and uniformity.
>
> Regarding Prop 3.2, the sparsity property holds for *any* globally fixed baseline, as the proof relies on the masking mechanism itself. However, we agree interaction values and compatibility scores can be sensitive to this choice.
>
> **Empirical Validation:** We conducted an ablation study across **all 6 datasets** (Vanilla LR, 10 seeds) comparing the **Mean** baseline against two semantically valid alternatives: **Median** (ensures integer values) and **Mode** (ensures valid categorical codes).
>
> As shown at `https://icml2026authorresponse-paper808.netlify.app/`, cosine similarity between metric decomposition vectors from different baselines averages **0.96** (Mean↔Median **0.969**, Mean↔Mode **0.944**, Median↔Mode **0.969**). This confirms that the **direction of coalition-level representations remains highly consistent**. Furthermore, the compatibility landscape (specifically Fairness-Utility near-orthogonality) is robustly preserved under both alternatives (see Figure at link).
>
> **Q1: Distributional Masking**
>
> Yes, generalizing to distributional masking (e.g., marginal sampling) would resolve the OOD issue by evaluating expected model responses over empirical distributions. We opted for the mean baseline primarily for **computational efficiency**, as distributional masking increases cost by a factor of $B$ (number of samples).
>
> Importantly, Ren et al. (2023, ICLR) theoretically proves that more faithful baselines yield sparser interactions, while Ren et al. (2024, ICLR) shows that DNNs naturally exhibit sparse interactions with low-order dominance. Since our framework identifies low-order coalitions as primary compatibility drivers (Sec 4.2), distributional masking (more faithful) would likely **strengthen** our core findings. Given our median/mode results showing landscape robustness, we expect distributional masking to further consolidate rather than alter the geometric structure. We will add this discussion and the OOD limitation to Appendix C.
>
> ---
>
> **Q2: Scalability and Feature Limits**
>
> Exact computation scales at $\mathcal{O}(2^{|N|})$. Without approximations, the realistic maximum is $|N| \le 15$ features.
>
> However, our framework is structurally agnostic to interaction computation methods and can directly integrate with recent approximation algorithms for higher-dimensional data, including:
> - Sparsity-regularized learning (Ren et al., 2023, CVPR; Ren et al., 2023, ICLR)
> - PCA-based dimensionality reduction (Ren et al., 2025, AAAI)
>
> We will expand the Limitations section to explicitly discuss this bottleneck and these scaling avenues.

---

> > ### Author Rebuttal · Reviewer_sZbz · 2026-04-03
> >
> > I would like to thank the authors for the clarification on W1 - it addressed my questions about the choice of cosine similarity. Also my questions about how it should work in discrete settings were answered. I also thank for the clarification regarding distributional masking, I think it would still benefit from empirically testing the prediction that authors made.

---

> > > ### Author Response · Authors · 2026-04-06
> > >
> > > We appreciate this constructive suggestion. Following your guidance, we conducted additional experiments using **marginal sampling** (K=100, stratified) as a distributional masking strategy. The results empirically validate our prediction.
> > >
> > > ### Experimental Setup
> > >
> > > **Baseline Strategy:** Marginal sampling with $K=100$ strata per feature. This increases computational cost by approximately $K$-fold compared to the mean baseline. Formally, for each subset $S$ of features, the reward is computed as:
> > >
> > > $$v\_{\text{marginal}}(S) = \mathbb{E}\_{x'\_{N\setminus S} \sim P(X\_{N\setminus S})}\left[f(x\_S, x'\_{N\setminus S})\right] \approx \frac{1}{K}\sum\_{k=1}^{K}f\left(x\_S, x^{\prime(k)}\_{N\setminus S}\right)$$
> > >
> > > where $x^{\prime(k)}\_{N\setminus S}$ are sampled independently from the training set for each masked feature. We employ stratified sampling (sorting values into $K$ percentiles) to reduce variance.
> > >
> > > **Configuration:**
> > >
> > > * Model: Vanilla Logistic Regression
> > >
> > > * Coverage: 6 datasets × 10 seeds = 60 configurations
> > >
> > > * Metrics: All 9 metrics (4 group fairness, 2 individual fairness, 3 utility)
> > >
> > > ### Key Results (\`<https://icml2026replyrebuttalcomment-paper808.netlify.app/>\`)
> > >
> > > **Compatibility Landscape (the boxplot):** The hierarchical structure is preserved:
> > >
> > > * Utility-Utility: strong alignment
> > >
> > > * Fairness-Fairness: synergy
> > >
> > > * Fairness-Utility: near-orthogonality
> > >
> > > **Interaction Sparsity (the violin plot):** Low-order dominance confirmed and strengthened:
> > >
> > > * $\[0, 0.3n]$: Dominant contributions with high variance
> > >
> > > * $(0.3n, n]$: Negligible contributions
> > >
> > > ### Interpretation
> > >
> > > 1. **Robustness validated:** Core findings (near-orthogonality, low-order dominance) are not artifacts of the mean baseline or other global baseline method.
> > >
> > > 2. **Prediction confirmed:** As hypothesized, more faithful baselines consolidate—rather than alter—the geometric structure.
> > >
> > > 3. **OOD concern addressed:** Marginal sampling ensures in-distribution masked values while preserving the compatibility landscape.
> > >
> > > ### Limitation
> > >
> > > Marginal sampling assumes feature independence and relies on computationally intensive sampling (compared to global baselines). Conditional sampling could provide more faithful representations but at substantially higher computational cost.
> > >
> > > ### Conclusion
> > >
> > > This experiment empirically validates our prediction that distributional masking would "strengthen our core findings" (Q1 Rebuttal). The compatibility landscape's robustness across multiple baseline strategies reinforces the reliability of our framework's conclusions. We will include this analysis in Appendix C of the revised manuscript.
> > >
> > > Thank you again for the feedback that helped refine this work.

---

### Decision · Program_Chairs · 2026-04-30

**Decision:**

Accept (regular)

**Comment:**

The paper investigates the relationship between fairness and utility in ML using a newly proposed game-theoretic framework; which decomposes each metric into Harsanyi interaction vectors and defines compatibility as the cosine similarity between them. This allows analyzing whether fairness and utility are misaligned.

All reviewers found this to be a novel a creative perspective that brings a different angle to understand fairness in ML. Some reviewers note that there are limitations such as not being able to handle all possible utilities, but overall the paper presents a useful and original contribution based on which further research can be developed; so I recommend acceptance.